# Reduction of carbon, alkalinity and nutrient fluxes in the southern Baltic Sea caused by dragging of otter trawl nets across the seafloor

- Pankan Linsy <sup>1</sup>, Stefan Sommer <sup>1</sup>, Jens Kallmeyer <sup>2</sup>, Simone Bernsee <sup>2</sup>, Florian Scholz <sup>3</sup>, Habeeb Thanveer Kalapurakkal <sup>1a</sup>, Andrew W. Dale <sup>1</sup>
  - <sup>1</sup> GEOMAR Helmholtz Centre for Ocean Research Kiel, Wischhofstraße 1–3, 24148 Kiel, Germany
  - <sup>2</sup>GFZ Helmholtz Centre for Geosciences, Telegrafenberg, 14473 Potsdam, Germany
  - <sup>3</sup> Institute for Geology, Center for Earth System Research and Sustainability, Universität Hamburg, Hamburg, Germany.
    - <sup>a</sup> Now at: Alfred Wegener Institute Helmholtz Centre for Polar and Marine Research, 27570 Bremerhaven, Germany

Correspondence to: Pankan Linsy (linsyp@geomar.de)

15

10

## **Abstract**

Mobile bottom-contacting fishing (MBCF) represents a substantial anthropogenic disturbance, significantly 20 disrupting seafloor integrity and altering oceanic carbon storage. In this study, we conducted a benthic trawling experiment on organic-rich muddy sediments in the Mecklenburger Bight, southern Baltic Sea, employing an otter trawl. Multiple trawl tracks were made to assess the temporal impact of bottom fishing on the benthic ecosystem over time scales ranging from days to weeks. Focus was on the wide area where the net footrope was dragged between the otter boards, rather than on much smaller area impacted by the trawl doors. This study constitutes the 25 first comprehensive investigation to systematically monitor the effects of MBCF on benthic oxygen, carbon, alkalinity and nutrient fluxes using autonomous in situ lander measurements. Seafloor observations revealed a profound impact of trawling on seafloor morphology. Flux measurements, coupled with sediment data, indicated reductions in benthic fluxes of O<sub>2</sub>, dissolved inorganic carbon (DIC), total alkalinity (TA), and nutrients (PO<sub>4</sub><sup>3-</sup>, NH<sub>4</sub><sup>+</sup>, and H<sub>4</sub>SiO<sub>4</sub>) within trawled areas compared to control sites. Additionally, observed decreases in organic 30 carbon remineralization rates suggest that MBCF alters benthic respiration by disrupting key biogeochemical processes. Fluxes of O2, DIC, and TA had not returned to baseline levels by the conclusion of the 16-day observation period, indicating prolonged disturbance effects, although natural temporal variations may have an influence. Despite substantial alterations to the benthic biogeochemical pathways, modeling suggests that the reduction in benthic DIC and TA fluxes exerts only a minor influence on CO2 release to the atmosphere compared 35 to the potential impact of pyrite oxidation in resuspended sediment.

# 1. Introduction



Mobile bottom-contacting fishing (MBCF) is one of the major anthropogenic activities that significantly affect the marine environment (Depestele et al., 2019; Hiddink et al., 2006; Oberle et al., 2016b; Olsgard et al., 2008; Pusceddu et al., 2014). Trawling involves catching benthic and demersal fish by towing nets along the seafloor using various devices. MBCFgear can penetrate the seafloor and dislocate the sediment to varying depths depending on the sediment type and gear used (Martín et al., 2014). Approximately 4.9 million km² or 1.3% of the global ocean is estimated to be trawled each year (Sala et al., 2021). The impacts of MBCF include the alteration of seabed morphology through the scraping, ploughing, and resuspension of surface sediments (Bruns et al., 2023; Oberle et al., 2016b). These processes ultimately disrupt benthic biogeochemical cycles (Allen and Clarke, 2007; Bradshaw et al., 2021; Bruns et al., 2023; Hale et al., 2017; Morys et al., 2021; Sciberras et al., 2016; Tiano et al., 2019; van de Velde et al., 2018). Additionally, MBCF has been shown to significantly reduce benthic fauna biomass (Bergman, 2000; Bergman and Meesters, 2020; Tiano et al., 2022) and, ultimately, change benthic community structures (Bradshaw et al., 2024; Kaiser et al., 2002; Pusceddu et al., 2014).

The seafloor disturbance caused by trawling has been argued to disrupt oceanic carbon sequestration (Atwood et al., 2024; Epstein et al., 2022; Hiddink et al., 2023; Sala et al., 2021; Zhang et al., 2024) as marine sediments serve as a critical reservoir for long-term carbon storage (Burdige, 2007; LaRowe et al., 2020). The study by Sala et al., (2021) suggests that seafloor disturbances caused by trawling and dredging result in the annual release of 0.58-1.47 Pg of CO<sub>2</sub>, largely due to increased remineralization of particulate organic carbon (POC). Atwood et 55 al. (2024) estimated that 50-60% of the CO<sub>2</sub> released due to trawling is emitted into the atmosphere over a decade, accounting for approximately 0.34-0.37 Pg CO<sub>2</sub> per year globally. In particular, trawling in shelf seas has been shown to reduce POC by 29% (Porz et al., 2024), with long-term losses equivalent to emissions of 3.67 Mg CO<sub>2</sub> km<sup>-2</sup> yr<sup>-1</sup>, assuming complete mineralization of the disturbed POC (Zhang et al., 2024). However, a subsequent metadata analysis by Epstein et al., (2022) yielded mixed findings on the impact of trawling on POC stock, 60 revealing no significant effect in 61% of the 49 studies analyzed. Among the remaining studies, 29% reported decreased POC levels associated with fishing activities, while 10% observed an increase in POC. These results indicate that the impact of MBCF is site-specific and probably influenced by factors such as trawling frequency, type of fishing gear, local lithology, and towing intensity (De Borger et al., 2021; Depestele et al., 2019; Oberle et al., 2016a; Tiano et al., 2019). Thus, site-specific studies are necessary for studying the impact of MBCF in any 65 given region (Stephens and McConnaughey, 2024).

Bottom fishing is carried out over large swathes of the Baltic Sea (HELCOM, 2018), and approximately 36% of the seafloor in the southwestern Baltic Sea is affected (Díaz-Mendoza et al., 2025). A study by Van Denderen et al. (2020) revealed that trawling, together with bottom water hypoxia, has led to a 50 % reduction in benthic biomass in 14% of the Baltic Sea region. The impact of MBCF on the benthos in the Baltic Sea has been the subject of numerous studies (Bradshaw et al., 2021, 2024; Bunke et al., 2019; Corell et al., 2023; Morys et al., 2021; Porz et al., 2023; Rooze et al., 2024; Schönke et al., 2022), but relatively few have examined the impact on benthic biogeochemistry (Bradshaw et al., 2021, 2024; Morys et al., 2021; Rooze et al., 2024; Tiano et al., 2024). Morys et al. (2021) conducted a controlled benthic dredging experiment in previously untrawled regions of the Baltic Sea. The experiment simulated the sediment disturbance caused by trawl gears by removing the surface sediment, allowing for a comparative analysis of geochemical characteristics with an adjacent pristine area. Immediate post-disturbance observations indicated alterations in benthic nutrient fluxes and porewater profiles. Bradshaw et al., (2021) conducted a controlled field experiment in muddy sediments of the Baltic Proper using a commercial otter trawl to quantify the physical and biogeochemical impacts of MBCF. Their findings indicated a reduction in oxygen penetration depth into the seafloor as well as altered nutrient and oxygen fluxes across the sediment-water interface. The trawl track persisted for at least 18 months, which suggests that sediment biogeochemistry may not fully recover in areas with frequent MBCF. However, as a single-track experiment, these findings may not fully represent the cumulative impacts of trawling, particularly in areas where MBCF occurs intensively more than 10 times per year (HELCOM, 2018).









When examining the impact of trawling activities on the oceanic carbon pool, it is also crucial to consider the effects of trawling on sedimentary total alkalinity (TA) and dissolved inorganic carbon (DIC) fluxes rather than solely focusing on organic carbon remineralization. Sediment-water reactions influence the ocean's buffering capacity, thereby affecting its ability to take up CO<sub>2</sub> from the atmosphere (Zeebe and Wolf-Gladrow, 2001). In the continental shelf ecosystem, in addition to the burial of organic carbon, benthic alkalinity production plays a key role in net carbon sequestration (Van Dam et al., 2022). The shallowness of coastal seas further permits a close interaction between sediments and the atmosphere (Brenner et al., 2016; Burt et al., 2016). Several processes control alkalinity generation in sediments and fluxes to the water column, such as mineral formation and dissolution, denitrification, pyrite formation and burial, and reverse weathering (Hu and Cai, 2011; Krumins et al., 2013; Middelburg et al., 2020). Among these, alkalinity production associated with pyrite formation likely constitutes a significant blue carbon sink (Hu and Cai, 2011; Reithmaier et al., 2021). Experimental and modeling studies have shown that trawling-induced resuspension reduces the capacity of the Baltic Sea to remove atmospheric CO<sub>2</sub> by decreasing alkalinity, mainly through the oxidation of pyrite (Kalapurakkal et al., 2025). More broadly, MBCF and dredging activities are estimated to reduce alkalinity generation, thereby weakening the marine carbon sink by approximately 2-8 Tg CO<sub>2</sub> per year, through their impact on both organic and inorganic carbon cycling (van de Velde et al., 2025). The field observations conducted to date have not specifically investigated the impact of MBCF on alkalinity fluxes. Therefore, additional field-based investigations are essential to comprehensively understand the influence of MBCF on the broader carbon cycle in addition to the associated biogeochemical upheavals at the sediment surface.

As a step toward addressing these critical knowledge gaps, we conducted a benthic trawling experiment in Mecklenburger Bight within the German Exclusive Economic Zone (EEZ) of the Baltic Sea. An otter trawl was deployed to create multiple trawl tracks. Subsequently, dissolved oxygen, DIC, TA, and nutrient fluxes were measured in situ using benthic landers to assess the effect of trawling on sediment regeneration rates, as well as a

detailed survey of sediment biogeochemical parameters in recovered sediment cores. Due to low probability of deploying large gears directly on the trawl marks gouged out of by the otter boards, we chose to target instead the much wider net area between the trawl boards (termed the 'net zone'). In addition to the benthic working program, we carried out seafloor imaging to observe the degree of disturbance and redistribution of sediments possibly affecting benthic communities. The major aim of the fieldwork was to determine the direct effects of MBCF on the benthic ecosystem on time scales of days to weeks and the regeneration capacity of sediment biogeochemistry. We also explored how changes to benthic TA and DIC fluxes impact air-sea CO<sub>2</sub> exchange. Our results provide novel insights into trawling-induced alterations in benthic biogeochemical processing in coastal environments.






110

## 2. Materials and Methods

## 2.1 Study area and experimental strategy

The research cruise AL616 on RV Alkor was undertaken from 18 July to 9 August 2024 in the German EEZ of the Baltic Sea (Sommer et al., 2025). The cruise was embedded in a larger campaign further addressing effects of trawling on different size classes of benthic communities (macrobenthos, meiofauna, protozoans and bacteria/archaea), including sedimentological investigations. Aside from RV Alkor, four research vessels participated in this trawling experiment. On RV Elisabeth Mann Borgese (cruise EMB 345), sediments were sampled for biological analyses. Scientists on the RV Clupea (cruise #389) carried out the trawling and investigated the effects of trawling on fish populations. SCUBA diving operations were performed using the RV Limanda, and the small tender Klaashahn was used to survey the seabed using hydroacoustics. Trawling operations were conducted using a standardized bottom trawl commonly employed in Baltic demersal surveys (TV-3#520x80 mesh size). The gear was equipped with Thyborøn Type 2 Standard trawl doors, each with a surface area of 1.78 m² (Fig. S1). The distance between the otter boards on either side of the trawl net was approximately 60 meters (ICES, 2017). Sediment disturbance within the net area results from interactions with the back strop, sweep lines, chains, bridles, foot rope, and fishing line (see Fig. S1).

The working area was located in the Mecklenburger Bight (southwestern Baltic Sea) 5 km offshore the town of Kühlungsborn at water depths of  $\sim$ 23 m (Fig. 1). It comprises the High-Impact (HI) area with dimensions of 1950 x 300 m that was trawled several times in an east – west direction on 20 July 2024. A Control area (CL) of similar size where no trawling was performed was located west of the HI area. The sediments across the working area were fine-grained muds.

**Figure 1:** Trawling area indicating BIGO (yellow squares), MUC (green circles), CTD (white circles), and deep-sea rover (DSR) (red circles) deployment sites. Red lines indicate trawl tracks. The inset shows the sampling area (red box) in the western Baltic.

## 2.2 Water and sediment sampling







A Sea-Bird Scientific CTD (SBE 9) equipped with a 12-Niskin bottle water sampling rosette was utilized for in situ measurements of salinity, temperature, depth and chlorophyll-a and for retrieving water samples. Data acquisition was performed using Sea-Bird Seasave software (v7.26.7). A total of seven CTD casts were conducted, with three deployments in the CL and HI areas and one outside. Water samples for dissolved oxygen measurement were immediately analyzed onboard using the Winkler method. Samples for nutrient analysis were collected and analyzed onboard.

Sediment samples were collected using a multiple-corer (MUC) with 6 and 11 deployments in the CL and HI areas, respectively (Table S1). Two cores (MUC1 and MUC2) were collected from the HI area prior to the trawling experiment and are considered control samples. The MUC was equipped with seven plastic liners, each 60 cm in length and 10 cm in internal diameter. The MUC was lowered toward the seafloor at a speed of 0.3 m s<sup>-1</sup> in every deployment. Upon reaching the seafloor, lead weights applied gravitational force to drive the liners into the sediment, with a maximum penetration depth of 35 cm. Upon retrieval, all cores were moved to a cooling laboratory maintained at 10°C, which approximates the bottom water temperature at the sampling sites. The cores were sectioned at a resolution of 1 cm near the surface, increasing to 4 cm for sediment depths greater than 20 cm. For all measurements and sub-sampling for redox-sensitive parameters from the MUC cores, the sediments were sectioned inside an argon-filled glove bag to prevent contact with atmospheric oxygen and filled into 50 ml Falcon tubes. Porewater was extracted by centrifuging the samples at 4000 rpm for 20 minutes at 3°C in a refrigerated centrifuge, effectively separating the porewater from the particulates. Following a mechanical failure of the centrifuge, porewater was obtained by inserting Rhizon filters (0.2 µm) into pre-drilled holes in the lids of the centrifuge tubes. Porewaters were then extracted under vacuum using 20 ml plastic syringes. All porewater samples were then filtered (0.2 µm cellulose-acetate syringe filters) under argon. Supernatant bottom water samples of the MUC cores were also collected at each station. Additional sediment samples cores were stored at 4°C in pre-weighed airtight containers for porosity and particulate geochemistry analyses.

For quantification of sulfate reduction rates, we used the radiotracer incubation technique of Jorgensen (1978). Three acrylic tubes (24 mm internal diameter, 3 mm wall thickness) were carefully inserted into a single MUC core from the same deployment as for the geochemistry analyses using suction to avoid compaction during insertion. The tubes were removed from the sediment, both ends sealed with rubber stoppers and stored in an incubator at approximate in-situ temperature. At the end of each day, <sup>35</sup>SO<sub>4</sub><sup>2-</sup> radiotracer solution was injected into the cores, from the water-sediment interface down to 20 cm sediment depth. The radiotracer (15 μl volume, activity ca. 200 kBq) was injected in 1 cm intervals through silicone-sealed holes with 2 mm diameter. The cores were then incubated for 24 h in the dark.

After the incubation, the sediment was pushed out of the acrylic tubes and sectioned. A resolution of 1 cm was selected for the first 6 cm, then 2 cm down to 10 cm, followed by 5 cm for subsequent depths down to 20 cm. Each sediment section was placed in a 50 ml centrifuge tube, pre-loaded with 10 ml of 20% zinc acetate solution to fix all hydrogen sulfide and terminate microbial activity. Samples were thoroughly shaken to break up sediment aggregates and then frozen overnight. Further storage and transport to the home lab was done at room temperature.

### 2.3 In situ flux measurements with BIGO landers











In situ flux measurements were conducted using two Biogeochemical Observatories (BIGO-I and BIGO-II). The BIGOs were deployed 3 and 7 times in the CL and HI areas, respectively. Each lander had two measurement chambers (C1 and C2) with a diameter of 28.8 cm and an area of 651 cm<sup>2</sup> (Sommer et al., 2009, 2016). Details of lander deployment protocols can be found elsewhere (Sommer et al., 2009, 2016). Sediments were incubated for 30 to 48 hours, and discrete samples (~47 mL) of overlying water were collected at pre-programmed intervals using glass syringes. The glass syringes were connected to the chambers through 1-meter-long Vygon tubes filled with distilled water with a dead volume of 6.9 mL. An additional eight glass syringe water samples were collected to monitor ambient bottom water conditions at the same time intervals. Another sample set was collected for the analysis of DIC and TA in quartz glass tubes. Water samples from the glass syringes were immediately transferred to a cooling laboratory and subsampled for analyses of nitrate (NO<sub>3</sub>-), nitrite (NO<sub>2</sub>-), ammonium (NH<sub>4</sub><sup>+</sup>), phosphate (PO<sub>4</sub><sup>3-</sup>), and silicic acid (H<sub>4</sub>SiO<sub>4</sub>). Oxygen was measured inside the chambers and in the ambient seawater using optodes (Aanderaa). They were two point calibrated before each lander deployment using well-oxygenated seawater and anoxic seawater, which was produced by adding 5 to 15 g of Na<sub>2</sub>S. The dilution error from the distilled water in the glass syringe samples, was corrected using chloride concentrations measured in the syringe samples and the ambient seawater. Nutrient and oxygen fluxes were determined by multiplying slope of the linear regression of the concentration versus time slope by the height of the water inside the chamber. Fluxes determined this way include fluxes by molecular diffusion in addition to non-local transport by bioirrigation and thus constitute the total solute flux. Oxygen fluxes are thus reported as total oxygen uptake (TOU), and as a positive number. Otherwise, reported positive fluxes are directed into the water column and vice versa. Separate BIGO sediment cores were collected by pushing short liners (10 cm in diameter, approximately 20 cm in length) into the sediment within the incubation chambers after BIGO retrieval. Sediment cores were subsampled, and porewater was extracted similarly to MUC cores. The BIGO incubation chambers were also subsampled for SRR measurements, directly inserting three acrylic tubes into each chamber after recovery of the lander. Samples were treated identical to those from MUC cores.

## 2.4 In situ flux O2 measurements with Deep-sea Rover (DSR) Panta Rhei

In addition to the fluxes measured using the BIGOs, the autonomous Deep-Sea Rover (DSR) Panta Rhei was used to repeatedly measure O<sub>2</sub> uptake in the HI and the CL. The DSR represents a six-wheeled vehicle (weight in air 1200 kg, 80 kg in water, dimensions 3 x 2 x 1.7 m (L, W, H)), which is specifically designed to carry out repeated benthic oxygen flux measurements inside two benthic incubation chambers at the front of the vehicle for prolonged time periods of up to one year.

O<sub>2</sub> was measured using fiber-optic O<sub>2</sub> sensors (Pyroscience, Bremen), three were placed inside each flux chamber, two sensors record the O<sub>2</sub> variability in the ambient bottom water. The volume of the two benthic chambers was either determined automatically by injecting pure water into the benthic chambers after the measurement and simultaneously recording the change of the conductivity (Aanderaa conductivity sensors) or by vertical centimeter scales mounted at the outside of each transparent chamber. From these data, the TOU was determined in same way as for the BIGOs. Additionally, the rover carried three camera systems; two of which were forward looking focusing on the sediment area where flux measurements are conducted by the left and right flux chamber, whereas the third camera was backward looking.

Upon its placement on the seafloor, the following sequence of activities was performed by the rover: i. after placement, it moved a distance of several meters away from its landing site into an undisturbed area, ii. the benthic chambers were flushed, iii. photos taken of the area that will be sampled by the two benthic chambers iv. benthic chambers inserted into the sediment and flushed prior to the start of flux measurement, v. photos taken of both chambers whilst inserted into the sediment (these photos can be used for the volume determination of each chamber), vi. measurement phase, duration was set to 8 hours, the sampling interval of the optodes was set to 5 min. vii. after the measurement phase, pure water was injected to determine the volume of both chambers, and the change of conductivity was monitored for 20 min at an increased sampling rate, viii. the chambers were raised out of the sediment and photos of the sampled area were again taken, ix. the rover advanced 0.7 m to the next measurement area. Apart from step i, this sequence was repeated until the recovery of the rover.

## 2.5 Geochemical analyses of sediments









Porewater samples were immediately analyzed for TA, dissolved nutrients (TPO<sub>4</sub><sup>3-</sup>, NO<sub>3</sub><sup>-</sup>, NO<sub>2</sub><sup>-</sup>, NH<sub>4</sub><sup>+</sup>, and H<sub>4</sub>SiO<sub>4</sub>), Fe<sup>2+</sup>, and dissolved hydrogen sulfide (H<sub>2</sub>S). TA was measured by titrating 0.5–1 mL of sample with 0.02 N HCl using a color indicator (mixture of methylene blue and methyl red), following the method of Ivanenkov and Lyakhin (1978). The endpoint was identified by the appearance of a stable pink color. During titration, the sample was degassed by continuous nitrogen bubbling to eliminate any CO<sub>2</sub> and H<sub>2</sub>S produced. Standardization was performed with the International Association for the Physical Sciences of the Oceans (IAPSO) seawater solution, achieving an analytical precision error for TA of better than 3%.

Porewater H<sub>2</sub>S, NH<sub>4</sub>+, PO<sub>4</sub><sup>3-</sup>, and H<sub>4</sub>SiO<sub>4</sub> concentrations were determined by standard spectrophotometric methods using a Hitachi U-2001 spectrophotometer (Grasshoff et al., 1999). NO<sub>3</sub><sup>-</sup> and NO<sub>2</sub><sup>-</sup> were analyzed with a Quaatro Autoanalyzer (Seal Analytic), with an error margin below 2%. Dissolved Fe<sup>2+</sup> concentrations were measured by taking 1 mL subsamples within a glove bag, stabilizing immediately with ascorbic acid, and analyzing within 30 minutes following complexation with 20 µl of ferrozine. The analytical error for Fe<sup>2+</sup> measurement was within ±5%. For H<sub>2</sub>S analysis, a porewater aliquot was diluted with oxygen-free artificial seawater, and sulfide was fixed by the immediate addition of zinc acetate gelatine solution.

Untreated filtered samples were stored for later SO<sub>4</sub><sup>2-</sup>, Cl<sup>-</sup>, and Br<sup>-</sup> analysis via ion chromatography at GEOMAR. Additional subsamples were acidified (20 µl of suprapure HNO<sub>3</sub> added to 2 mL sample) for the analysis of major ions (K, Li, B, Mg, Ca, Sr, Mn, Br, and I) and trace elements by inductively coupled plasma optical emission spectroscopy (ICP-OES). Nutrient concentrations in water samples collected from landers were analyzed using the autoanalyzer, with analytical precision and detection limits reported by Haffert et al., (2013) and Sommer et al. (2025).

The water content of sediment cores was determined at by calculating the difference between the wet and dry weights of the samples. Porosity was then calculated from the water content, assuming a particle density of 2.5 g cm<sup>-3</sup> and a seawater density of 1.023 g cm<sup>-3</sup>. After freeze-drying, samples were ground using ball mills. Dry sediment was used for analysis of total carbon, nitrogen (assumed to be particulate organic N, PON) and total sulfur (TS) using a Euro elemental analyzer. POC content was determined after acidifying the sample with HCl (0.25 N) to transform the inorganic carbon to CO<sub>2</sub>. Particulate inorganic carbon, assumed to be calcium carbonate (CaCO<sub>3</sub>), was determined by weight difference between total and organic carbon. The precision and detection limit of the POC analysis was 0.04 and 0.05 dry weight percent (% C), respectively, while that for CaCO<sub>3</sub> was 2 and 0.1 % C.

The pyrite content of surface sediments (0-1 cm) was estimated by chromium-reducible sulfur following (Canfield et al., 1986). Freeze-dried and ground samples were used. Liberated sulfur was trapped as zinc sulfide and analyzed by photometry (Cline, 1969). The long-term reproducibility and accuracy of the method were monitored against analysis of pure pyrite mixed with quartz sand and an in-house standard (OMZ-2, Peru margin sediment).

Sulfate reduction rates (SRR) were quantified at GFZ Potsdam, using the cold chromium distillation technique (Kallmeyer et al., 2004). The sample vials were centrifuged and the supernatant carefully removed. A small aliquot of supernatant was kept for quantification of total radioactivity and the rest was discarded. The sediment pellet was quantitatively transferred into a distillation flask by flushing it out of the centrifuge vial with a total of 15 mL N, N-dimethylformamide (DMF), technical grade. To ensure complete mixing of the sediment with the chemicals a magnetic stir bar was added to the reaction flask. The flask was then connected to a constant stream of nitrogen gas (approximately 5-10 bubbles per second) in order to maintain strictly anaerobic conditions during the distillation. After 10 minutes of bubbling the DMF-sediment suspension with N<sub>2</sub> gas, 8 mL of 6 M hydrochloric

acid (HCl) and 15 mL of 1 M chromium (II) chloride solution were added to the flask via a reagent port. The chemicals will convert all reduced sulfur species (TRIS) to H<sub>2</sub>S, which was driven out of solution by the stream of nitrogen gas. The produced gas was then led from the flask through to a first trap filled with 7 mL of a buffered citric acid solution (19.3 g citric acid, 4 g NaOH in 1 L H<sub>2</sub>O, pH 4) to trap any aerosols potentially containing unreacted <sup>35</sup>SO<sub>4</sub><sup>2-</sup> radiotracer but let all H<sub>2</sub>S pass. Finally, the gas was bubbled through 7 mL of 5% (w/v) zinc
acetate solution to trap all H<sub>2</sub>S as ZnS. To prevent overflowing of the zinc acetate trap, a few drops of silicon-based antifoam were added. After 2 hours of distillation, the contents of the zinc acetate trap, containing the produced H<sub>2</sub><sup>35</sup>S were quantitatively transferred into a 20 mL plastic scintillation vial and mixed with 8 mL scintillation cocktail for quantification of radioactivity by scintillation counting.

Sulfate reduction rates were calculated according to the following formula:

$$SRR = \frac{so_4 * \varphi * a_{tris} * 1.06 * 10^6}{a_{tot} * t}$$
Eq. (1)

where SRR is the sulfate reduction rate (pmol cm<sup>-3</sup> d<sup>-1</sup>); SO<sub>4</sub> is the sulfate concentration in the pore water (mmol L<sup>-1</sup>);  $\varphi$  is porosity, set to 0.7;  $a_{tris}$  is the radioactivity of total reduced inorganic sulfur (cpm);  $a_{tot}$  is the total radioactivity used (cpm); t is the incubation time (d); 1.06 is a correction factor for isotopic fractionation; and the factor  $10^6$  converts from  $\mu$ mol L<sup>-1</sup> to pmol L<sup>-1</sup>.

# 2.6 TA and DIC analysis of water samples




Samples for TA and DIC analysis were taken according to the slightly modified Standard Operation Procedures described by Dickson et al. (2007). The water samples obtained in the glass tubes were transferred into Pyrex test tubes (volume: 10 mL). Subsequently, a small headspace was applied, removing 0.3 mL of the water sample, and the samples were poisoned with  $10 \mu l$  of HgCl<sub>2</sub>. The test tubes were closed using greased glass stoppers (Apiezon vacuum grease) secured using plastic clamps, and then stored.

At GEOMAR laboratories, TA (Dickson, 1981) was titrated automatically in an open measuring cell (Metrohm, volume of measuring cell: 1 to 50 mL) according to Dickson et al. (2007). The sample volume was 1 mL. Hydrochloric acid (0.01 M) was added stepwise and the pH value was continuously recorded using an electrode (Metrohm). The TA concentration was calculated using a modified Gran method (Gran, 1952; Humphreys, 2015). The precision and the deviation from the reference standard were 0.2 and 0.02 %, respectively. The CRM seawater standard Dickson batch #188 was used as the reference standard (total inorganic carbon content: 2099.26 µmol kg¹; total alkalinity: 2264.96 µmol kg¹, https://www.ncei.noaa.gov/access/ocean-carbon-acidification-data-system /oceans/Dickson CRM/batches.html).

The concentration of DIC was measured according to Dickson et al. (2007) using an Apollo SciTech analyzer (Model AS-C5) equipped with an infrared carbon dioxide (CO<sub>2</sub>) detector (Trace Gas Analyzer 7815-01, LICOR). The sample (0.6 mL) was mixed with 0.7 mL H<sub>3</sub>PO<sub>4</sub> in an extraction chamber where the DIC is completely converted to CO<sub>2</sub> and measured by infrared spectroscopy. A dilution series of a sodium carbonate standard (Na<sub>2</sub>CO<sub>3</sub>, 100 mM) was prepared for calibration, whereby salt water (7 g l<sup>-1</sup> NaCl) was used for the dilution. The CRM standard (batch 188) served as the reference standard. The precision and the deviation from the standard were 0.2 and 0.3 %, respectively.

## 2.7 Diffusive flux calculations

The diffusive flux of dissolved constituents in porewater between the bottom water and surface sediment was calculated according to Fick's law:

$$J_{C} = \phi \times D_{s} \times (dC/dz)$$
 Eq. (2)

where J<sub>C</sub> represents diffusive flux of solute C at the sediment-water interface, φ is porosity, D<sub>s</sub> is the diffusion coefficient in sediments, and dC/dz is the concentration gradient at the sediment-water interface. The diffusion coefficient was derived from the diffusion coefficient in seawater (D<sub>sw</sub>) at in situ temperature and salinity using the following equation (Li and Gregory, 1974):

$$D_s = D_{sw}/\theta^2$$
 Eq. (3)

where  $\theta^2$  accounts for sediment tortuosity calculated according to Boudreau (1997):

$$\theta^2 = 1 - \ln(\phi)^2$$
 Eq. (4)

The diffusion coefficient for TA was assumed to be equal to that for bicarbonate ion (Dale et al., 2021a). Concentrations of TA, NH<sub>4</sub><sup>+</sup>, and H<sub>4</sub>SiO<sub>4</sub> gradually increased with sediment depth, which allowed the concentration gradient to be calculated by taking the derivative at the sediment-water interface of an exponential function fitted through the concentration data in the upper 4 cm:

$$C(z) = \alpha - (\alpha - C(0)) \times e^{-\gamma z}$$
 Eq. (5)

C(z) – concentration at depth z

α – asymptotic concentration at 4 cm depth

C(0) – bottom water concentration

γ – exponential coefficient

z – sediment depth

The best fit to the data was determined using the FindFit function in Mathematica software. Concentrations of Fe<sup>2+</sup>, NO<sub>3</sub><sup>-</sup>, NO<sub>2</sub><sup>-</sup>, H<sub>2</sub>S and PO<sub>4</sub><sup>3-</sup> were more scattered and not easily represented by Eq. (5), and their diffusive fluxes were not calculated. To assess variability in porewater fluxes across datasets and to facilitate comparison with the BIGO fluxes, we computed the average flux per site and visualized the distribution using box-and-whisker plots. This descriptive approach provides a visual representation of the spread in the data, facilitating comparisons between the HI and CL areas.


## 2.8 POC degradation rate calculations

The rates of particulate organic carbon degradation (RPOC<sub>TOT</sub>) were estimated using a mass balance approach based on the measured fluxes (F) of O<sub>2</sub>, NO<sub>3</sub>, and NH<sub>4</sub><sup>+</sup> (Dale et al., 2014):

$$RPOC_{TOT} = \frac{r_{CN}(2 F_{NH4} r_{NO3} - F_{O2} r_{NO3} - F_{NH4} r_{O2} - F_{NO3} r_{O2})}{2 r_{NO3} - r_{O2} + r_{CN} r_{NO3} r_{O2}}$$
 Eq. (6)

where  $r_{CN}$  represents the atomic ratio of carbon-to-nitrogen (C:N) ratio in organic matter undergoing degradation, calculated from the mean down-core POC and PON contents.  $r_{O2}$  and  $r_{NO3}$  were 1 and 0.8, respectively, assuming organic matter with an oxidation state of zero.



## 2.9 Statistical analyses

Statistical analyses were conducted using the R software package. The non-parametric Mann-Whitney U test was employed for each set of data to assess the variability of fluxes between the HI and CL areas. Additionally, this test was also utilized to understand the difference in porewater profiles and variability in the particulate data between HI and CL areas. p-values <0.05 were considered indicative of statistically significant variability. The following notation is used: \*p < 0.05, \*\*p < 0.01, \*\*\*p < 0.001.

Linear mixed-effects models were used to assess the natural variability of fluxes over the three-week period. For each flux, models were fitted in R using the lme4 and lmerTest packages, with Area as a fixed effect and Date as

a random intercept to account for temporal variability. Models were estimated by maximum likelihood, and temporal effects were assessed using likelihood ratio tests comparing models with and without the Date random effect. Temporal variability was quantified from the Date random-effect variance, with higher values indicating greater temporal variability. *p*-values < 0.05 were considered statistically significant.

#### 3. Results




# 3.1 Environmental conditions at Mecklenburger Bight

Environmental conditions were monitored during the experiment from 18 July to 9 August 2024 using shipboard measurements of wind speed, wind direction, and surface water temperature (Fig. 2). Additionally, bottom water temperature and dissolved oxygen levels were measured during the different deployments of the DSR. Wind speed varied between 4.4 and 13.7 m s<sup>-1</sup>, with the highest gusts from a northwesterly direction. Correspondingly, sea surface temperature (SST) generally ranged between 18 and 20 °C with an increase to 21 °C by 1 August. The bottom water temperature, initially recorded at 9.8 °C, rose to 10.6 °C over the study period. Between 18 July and 2 August, bottom water  $O_2$  levels declined from approximately 100  $\mu$ mol  $L^{-1}$  to a minimum of 46  $\mu$ mol  $L^{-1}$ . Notable fluctuations in  $O_2$  concentrations were recorded between 31 July and 2 August, which coincided with reduced wind speeds and northerly winds. These variations were likely influenced by wind speed oscillations between 0.5 and 6 m s<sup>-1</sup>, inducing seiches along the sloping seabed, where water masses of different  $O_2$  concentrations flowed past the rover.

A cyanobacterial bloom appeared in the study area between 3 and 6 August. During this period, SST ranged from 20.1 to 21.4 °C. In contrast, bottom water temperature remained decoupled from diurnal surface temperature fluctuations, gradually increasing from 9.7 to 10.3 °C. Simultaneously, bottom water  $O_2$  concentrations increased from 34 to 94  $\mu$ mol  $L^{-1}$ . The variations in bottom water temperature and  $O_2$  levels may be influenced by wind forcing, particularly a shift in wind direction from strong alongshore westerly winds to moderate southeasterly winds.

**Figure 2:** Environmental conditions during the time course of the trawling experiment including (a) air pressure, (b) wind speed (blue), wind speed v (north-south) component and direction (black), with arrows indicating orientation, (c) SST (red) and bottom water temperature (black), (d) bottom water dissolved O<sub>2</sub> concentration. Grey bars indicate periods where the ship stayed in harbor. The arrows above the figure denote the CTD deployments.

Initially, the thermocline and halocline were located at around 10 m depth and later shifted downwards due to the increase in wind velocity and mixing of the surface layer (Fig. 3). This shift was accompanied by a corresponding deepening of the nutricline. Over time, the chlorophyll-a maxima intensified to almost 15 μg L<sup>-1</sup> when the cyanobacterial bloom intensified. In the surface mixed layer, O<sub>2</sub> concentrations were around 300 μmol L<sup>-1</sup> and

declined to  $\sim$ 110 µmol L<sup>-1</sup> at 4 m above the seafloor. Note that the low O<sub>2</sub> concentrations <100 µmol L<sup>-1</sup> recorded by the DSR (Fig. 2) at a distance of  $\sim$ 70 cm above the sediment were not detected by the CTD (maximum depth 5 m above the seafloor).

Figure 3: Water column distributions of profiles of temperature, salinity, chlorophyll-a, nutrients and oxygen.

Note that technical issues were observed with the pumps of CTD03 and CTD04, affecting temperature, salinity, and O<sub>2</sub> in the upper 12 m.

# 3.2 Trawling disturbance

415

420

405

The disturbance caused by the otter boards and footrope, sweeps, chains/, and bridles (compare Figure S1) on the seafloor morphology is distinctly visible in Fig. 4. The images show the undisturbed seabed on the left, characterized by a thin layer of brown phytodetritus. Moving to the right, a sharply defined furrow created by the left trawl board exposed the underlying dark-grey anoxic sediment. A narrow strip of relatively undisturbed sediment is observable adjacent to the furrow, followed by sediment casts on the inward side caused by the otter boards. Further to the right of the sediment casts, another narrow zone of minimally disturbed sediment can be observed. Finally, at the far right of the image, the sediment surface in the net zone has been visibly scraped off by the back strop and sweep line dragging across the seafloor during bottom fishing. A large sediment plume was also observed on the camera system immediately after trawling (not shown).

Figure 4: Ocean Floor Observation System (XOFOS) images taken across a trawl mark caused by the left otter board and the net area affected by the back strop and sweep linking the otter boards to the net (compare Fig. S1). The distance between the outer two green laser points is 45 cm. During trawling, the trawl door penetrates the sediment surface, displacing material from the furrow and depositing it towards the inner side. The area in between the otter boards is affected by the back strop, sweep lines, chains, bridles and the footrope and fishing line of the net. This area (referred to as net area) was targeted for sampling in the HI area. Details of the XOFOS deployments are provided in Sommer et al. (2025).

# 3.3 Sediment biogeochemistry and fluxes

435 Mean contents of POC, PON, CaCO3 and TS are presented in Fig. 5 and Table S2, along with organic C:N ratios and porosity. The sediment can be classified as organic-rich and carbonate-poor. POC, PON and CaCO<sub>3</sub> all decreased markedly in the upper 10 cm, and then stabilized at greater depth. TS, in contrast, accumulated to ~ 1 wt. % S. C:N ratios increased from ~ 8 to 9, indicating preferential remineralization of PON relative to POC. All particulate species in the HI area exhibited a slight reduction in the top 2-3 cm compared to the CL area. The 440 surface concentrations of POC, PON, and CaCO<sub>3</sub> were consistently higher at the CL sites (POC: 3.58 ± 0.74 wt.%; PON:  $0.54 \pm 0.11$  wt.%; CaCO<sub>3</sub>:  $5.39 \pm 1.12$  wt.%) compared to the HI sites (POC:  $2.77 \pm 0.78$  wt.%; PON:  $0.41 \pm 0.12$  wt.%; CaCO<sub>3</sub>:  $3.55 \pm 0.48$  wt.%). The reduction of POC in the HI area was close to 1 wt.%. These differences are statistically significant in the upper 1.5 cm, which suggest that surface sediment removal in the net area amounted to the 1-2 cm. This is confirmed by the porosity data, whereby the HI data overlap the CL data 445 when shifted downward by 1.5 cm (Fig. 5f). In the surface sediment layer, the concentrations of TS and C/N ratios showed no significant differences between the CL (TS:  $0.62 \pm 0.13$  wt.%; C/N:  $7.75 \pm 0.26$ ) and HI (TS:  $0.49 \pm 0.13$  wt.%; C/N:  $7.75 \pm 0.26$ ) and HI (TS:  $0.49 \pm 0.13$  wt.%; C/N:  $7.75 \pm 0.26$ ) and HI (TS:  $0.49 \pm 0.13$  wt.%; C/N:  $7.75 \pm 0.26$ ) and HI (TS:  $0.49 \pm 0.13$  wt.%; C/N:  $7.75 \pm 0.26$ ) and HI (TS:  $0.49 \pm 0.13$  wt.%; C/N:  $7.75 \pm 0.26$ ) and HI (TS:  $0.49 \pm 0.13$  wt.%; C/N:  $7.75 \pm 0.26$ ) and HI (TS:  $0.49 \pm 0.13$  wt.%; C/N:  $7.75 \pm 0.26$ ) and HI (TS:  $0.49 \pm 0.13$  wt.%; C/N:  $7.75 \pm 0.26$ ) and HI (TS:  $0.49 \pm 0.13$  wt.%; C/N:  $7.75 \pm 0.26$ ) and HI (TS:  $0.49 \pm 0.13$  wt.%; C/N:  $7.75 \pm 0.26$ ) and HI (TS:  $0.49 \pm 0.13$  wt.%; C/N:  $7.75 \pm 0.26$ ) and HI (TS:  $0.49 \pm 0.13$  wt.%; C/N:  $7.75 \pm 0.26$ ) and HI (TS:  $0.49 \pm 0.13$  wt.%; C/N:  $7.75 \pm 0.26$ ) and HI (TS:  $0.49 \pm 0.13$  wt.%; C/N:  $7.75 \pm 0.26$ ) and HI (TS:  $0.49 \pm 0.13$  wt.%; C/N:  $7.75 \pm 0.26$ ) and HI (TS:  $0.49 \pm 0.13$  wt.%; C/N:  $7.75 \pm 0.26$ ) and HI (TS:  $0.49 \pm 0.13$  wt.%; C/N:  $7.75 \pm 0.26$ ) and HI (TS:  $0.49 \pm 0.13$  wt.%; C/N:  $7.75 \pm 0.26$ ) and HI (TS:  $0.49 \pm 0.13$  wt.%; C/N:  $7.75 \pm 0.26$ ) and HI (TS:  $0.49 \pm 0.13$  wt.%; C/N:  $7.75 \pm 0.26$ ) and HI (TS:  $0.49 \pm 0.13$  wt.%; C/N:  $7.75 \pm 0.26$ ) and HI (TS:  $0.49 \pm 0.13$  wt.%; C/N:  $7.75 \pm 0.26$ ) and HI (TS:  $0.49 \pm 0.13$  wt.%; C/N:  $7.75 \pm 0.26$ ) and HI (TS:  $0.49 \pm 0.13$  wt.%; C/N:  $7.75 \pm 0.26$ ) and HI (TS:  $0.49 \pm 0.13$  wt.%; C/N:  $7.75 \pm 0.26$  wt 0.17 wt.%; C/N:  $7.93 \pm 0.38$ ) sites.

Surface pyrite contents in the CL area  $(0.45\pm0.08 \text{ wt.\% FeS}_2)$  were similar to those of HI area  $(0.44\pm0.17 \text{ wt.\% FeS}_2)$ .

**Figure 5:** Solid-phase profiles at the study site. Mean values and standard deviations are shown for control sites (green) and high-impact areas (orange). The black points in the porosity profile represent the depth-adjusted values of the HI site assuming surface erosion of 1.5 cm.

Mean solute concentrations measured in MUC samples from CL and HI sites exhibited similar trends (Table S3, Fig. 6). The data have been normalized to chloride concentrations to account for down-core changes in salinity (Fig. 6i). Concentrations of TA, NH<sub>4</sub>+, PO<sub>4</sub><sup>3-</sup> and H<sub>4</sub>SiO<sub>4</sub> increased with depth due to the remineralization of organic matter. Concurrently, SO<sub>4</sub><sup>2-</sup> decreased by ~4 mM in the upper 10 cm and then showed little further decrease with depth. Concentrations of H<sub>2</sub>S were low in the upper 5 cm where a pronounced Fe<sup>2+</sup> peak was observed. H<sub>2</sub>S then increased to ~200 μM at the bottom of the cores. NO<sub>3</sub>- concentrations were elevated in the top 2 cm and then remained at levels close to the detection limit further down.

Sulfate reduction rates in these organic-rich sediments were elevated very close to the sediment surface and then decreased quasi-exponentially down to  $\sim 15$  cm. The depth-integrated SRR equaled  $4.1\pm3.1$  and  $3.1\pm2.7$  mmol S m<sup>-2</sup> d<sup>-1</sup> at the CL and HI sites, respectively. For a simple carbohydrate (i.e. zero C oxidation state), 2 moles of POC are oxidized per mole of  $SO_4^{2-}$  reduced (e.g., Burdige and Komada, 2011). The SRR thus translates to a POC oxidation rate of 8.3 and 9.5 mmol m<sup>-2</sup> d<sup>-1</sup> at the CL and HI sites.

Concentrations of TA, NH<sub>4</sub><sup>+</sup>, PO<sub>4</sub><sup>3-</sup>, H<sub>4</sub>SiO<sub>4</sub> and Fe<sup>2+</sup> tended to be higher in the upper cm in the CL area compared to the HI area. Conversely, SO<sub>4</sub><sup>2-</sup> tended to be lower, whereas H<sub>2</sub>S showed little difference. However, the variability of solute concentrations between CL and HI were not statistically significant.

**Figure 6:** Solute concentrations in porewaters at the study site and SRR rates. Mean concentrations and standard deviations are shown for control sites (green) and high-impact areas (orange).





Total fluxes (BIGO data) and diffusive fluxes (for TA, NH<sub>4</sub><sup>+</sup>, and H<sub>4</sub>SiO<sub>4</sub>) are illustrated in Fig. 7 (and Table S4). For TA and NH<sub>4</sub><sup>+</sup>, the two approaches of flux calculation were consistent with regards to the range and magnitude of the fluxes. H<sub>4</sub>SiO<sub>4</sub> fluxes showed larger differences with lower total fluxes from the BIGO. In general, the sediments were a sink for O<sub>2</sub> and NO<sub>3</sub><sup>-</sup> and a source for all other variables. NO<sub>2</sub><sup>-</sup> effluxes (not shown) were < 0.05 mmol m<sup>-2</sup> d<sup>-1</sup>. Mean TOU in the control area was 8.9 mmol m<sup>-2</sup> d<sup>-1</sup>, whereas DIC fluxes were much higher (mean 19.5 mmol m<sup>-2</sup> d<sup>-1</sup>), implying a significant contribution of carbonate dissolution to the DIC flux. This is also supported by a high mean TA flux of 12.2 mmol m<sup>-2</sup> d<sup>-1</sup> in the CL area. Following trawling, the fluxes of TOU, TA, DIC, NH<sub>4</sub><sup>+</sup>, PO<sub>4</sub><sup>3</sup>-, H<sub>4</sub>SiO<sub>4</sub> all declined compared to the CL area, whereas NO<sub>3</sub><sup>-</sup> and NO<sub>2</sub><sup>-</sup> fluxes showed little change. Notably, mean total TA and DIC fluxes were much lower in the HI area, falling to 5.4 and 10.2 mmol m<sup>-2</sup> d<sup>-1</sup>, respectively. The lower total fluxes of TA, NH<sub>4</sub><sup>+</sup>, PO<sub>4</sub><sup>3-</sup> and H<sub>4</sub>SiO<sub>4</sub> following trawling are further consistent with the lower porewater concentrations. Total fluxes of TA (\*\*p value), DIC (\*p value), NH<sub>4</sub><sup>+</sup> (\*p value), PO<sub>4</sub><sup>3-</sup> (\*p value), and H<sub>4</sub>SiO<sub>4</sub> (\*\*p value) exhibited statistically significant differences between the HI and CL areas. The diffusive fluxes of TA and H<sub>4</sub>SiO<sub>4</sub> were not significant, whereas NH<sub>4</sub><sup>+</sup> (\*p value) showed significant differences between CL and HI areas.

Control area estimates of RPOC<sub>TOT</sub> equaled 9.6 mmol m<sup>-2</sup> d<sup>-1</sup>, and were followed by a statistically significant (\*p value) decrease after trawling to 7.8 mmol m<sup>-2</sup> d<sup>-1</sup> (Fig. 7h). These are very similar to the TOU, which provides confidence in the RPOC<sub>TOT</sub> calculation. They are also similar to the depth-integrated SRR calculated above. Sulfate reduction is thus the major carbon remineralization pathway, as expected for organic-rich coastal sediments (e.g. Jørgensen, 2021; van de Velde et al., 2018).

Over the 16-day observation period in the HI area, the BIGO fluxes exhibited temporal variability, likely reflecting the gradual recovery of environmental conditions post-trawling (Fig. 8). TA and DIC effluxes and TOU apparently did not fully recover to the CL values by the end of the sampling program. The linear mixed-effects model analysis indicated that TOU, TA, and DIC exhibited natural temporal variability (Date random-effect variance > 0). Temporal trends in the fluxes of NH<sub>4</sub><sup>+</sup>, PO<sub>4</sub><sup>3-</sup>, and H<sub>4</sub>SiO<sub>4</sub> were less clear, although there is a tendency for higher

fluxes in the CL area versus the HI area. No significant natural temporal variability was detected for the other parameters (Date random-effect variance  $\sim$  0).

**Figure 7:** (a-g) Box-whisker plots showing solutes fluxes across the sediment-water interface calculated from lander data (BIGO) and porewater profiles (PW) at the control (green) and high-impact sites (orange). Mean fluxes are shown by the small circles. (h) POC remineralization rates were calculated using Eq. 5.

**Figure 8:** Plots showing the temporal evolution of total fluxes of solutes across the sediment-water interface (in mmol  $m^{-2} d^{-1}$ ) over 16 days in control (green) and high-impact (orange) areas determined from the BIGO data. Dashed horizontal lines indicate zero flux.

# 4. Discussion






# 4.1 Changes in TOU and nutrient fluxes in the net zone

The seafloor photographic images strikingly demonstrate the impact of trawling on sediment morphology (Fig. 4). Sediment was unevenly tossed into the net area by the otter boards, as evident in previous benthic images and model simulations (Bradshaw et al., 2021; Ivanović et al., 2011). In this study, we provide a comprehensive data set on biogeochemical fluxes based on in situ measurements using benthic landers, including DIC and TA fluxes. Focus is on the wide net area, which is far larger than that disturbed by the otter boards (1 - 2 m). Besides, for practical reasons, it was not possible to target the narrow mound and furrows gauged out by the otter boards with our large lander and coring equipment. The trawl marks were sampled by SCUBA divers using push cores, and this data set will be published in a forthcoming manuscript.

An important finding of the study relates to the total fluxes calculated using BIGO data and diffusive fluxes calculated using porewater gradients. The two approaches exhibited slight differences in magnitude for TA and NH<sub>4</sub><sup>+</sup>, whereas total H<sub>4</sub>SiO<sub>4</sub> fluxes were notably lower (Fig. 7). This mismatch could be of some concern since biogenic silica recycling efficiencies are often calculated using porewater concentration gradients (Dale et al., 2021b). Diffusive fluxes derived from concentrations measured in 1-cm intervals fail to capture the porewater gradients just below the sediment surface and do not account for non-diffusive fluxes arising from bioirrigation (Apell et al., 2018). In contrast, the fluxes obtained from benthic chambers integrate all the net sources and sinks across the sediment surface (Tengberg et al., 1995). Since porewater concentrations of H<sub>4</sub>SiO<sub>4</sub> were higher than in the bottom water, bioirrigation would tend to flush out H<sub>4</sub>SiO<sub>4</sub> and augment the total flux. A large sink for H<sub>4</sub>SiO<sub>4</sub> in the upper centimeter could account for the lower BIGO fluxes, although we find this hard to envisage given current knowledge about silica cycling in marine sediments (DeMaster, 2003). The lower benthic chamber fluxes thus suggest that the curve fitting procedure overestimates the H<sub>4</sub>SiO<sub>4</sub> flux at Mecklenburger Bight, but not for TA and NH<sub>4</sub><sup>+</sup>.

Figure 9: TOU versus bottom water oxygen concentration measured by the rover in control (green) and high-impact (orange) areas.

Oxygen concentrations in surface sediments are primarily regulated by the metabolic activities of micro-, meio-, and macrofauna, as well as the oxidation of reduced substances such as NH<sub>4</sub>+, Fe<sup>2+</sup> and H<sub>2</sub>S (Glud, 2008). TOU is further regulated by the content of reactive POC and the bottom water O<sub>2</sub> concentrations. At concentrations < 125 μM, characteristic of our study site, TOU is predicted to correlate positively with bottom water O<sub>2</sub> concentration (Cai and Reimers, 1995). Above 125 μM, the amount of organic carbon in the sediment becomes the critical factor. Our data show this behavior clearly (Fig. 9). The decline in bottom water O<sub>2</sub> concentrations over the study period (Fig. 2) proves to be an important control on TOU. Previous research has reported that trawling lowers O<sub>2</sub> consumption rates in sedimentary ecosystems by effectively stripping away surface sediments and associated microbial and faunal communities and by diminishing the availability of labile reduced substances (Allen and Clarke, 2007; Tiano et al., 2019; Bradshaw et al., 2024). Our findings suggest that trawling has a relatively minor impact on TOU flux in the HI area compared to the role of bottom water O<sub>2</sub> depletion. Similar observations have been reported in previous studies (Goldberg et al., 2014; Trimmer et al., 2005; Warnken et al., 2003), which also found limited effects of trawling on TOU. However, these studies did not explicitly assess the influence of declining bottom-water oxygen concentrations. We ascribe this feature to transport limitation of O<sub>2</sub> across the diffusive boundary layer when O<sub>2</sub> levels fall over time (Bouldin, 1968; Middelburg and Levin, 2009).

The reduction in benthic NH<sub>4</sub><sup>+</sup> fluxes post-trawling contrasts with Morys et al. (2021), who reported an increase immediately afterwards. They attributed this to NH<sub>4</sub><sup>+</sup>-enriched porewater at the newly formed sediment-water interface and the subsequent diffusion of NH<sub>4</sub><sup>+</sup> to the bottom water. Similarly, Duplisea et al. (2001) reported that the NH<sub>4</sub><sup>+</sup> flux from trawled sediment was 45 times higher than the flux from undisturbed sediments. Our findings align more with Bradshaw et al. (2021), who observed a decrease in NH<sub>4</sub><sup>+</sup> flux and argued that it was caused by NH<sub>4</sub><sup>+</sup> depletion following surface sediment removal or from a deceleration of biogeochemical turnover rates. Almroth-Rosell et al. (2012) also found that sediment resuspension resulted in decreased NH<sub>4</sub><sup>+</sup> efflux (~50%) in a benthic chamber investigation, which they ascribed to oxidation of NH<sub>4</sub><sup>+</sup> stimulated by nitrification (Almroth-Rosell et al., 2012). We argue for a simpler explanation for our site, whereby the lower NH<sub>4</sub><sup>+</sup> fluxes were caused

by removal of the reactive surface layer associated with the highest rates of ammonification; a finding unambiguously borne out by the reduction in DIC and TA fluxes, as well as by diagenetic modelling (De Borger et al., 2021). The same reasoning applies to the reduction in H<sub>4</sub>SiO<sub>4</sub> fluxes. The dissolution of biogenic silica, mainly diatoms in the Baltic Sea, is the main source of silicate in porewaters (Tréguer and De La Rocha, 2013). Given that dissolution rates are highest at the sediment surface where opal undersaturation is greatest (Rabouille et al., 1997), it logically follows that removal of this layer will immediately diminish the benthic H<sub>4</sub>SiO<sub>4</sub> flux.

Mean fluxes of NO<sub>3</sub><sup>-</sup> into the sediment (0.3 – 0.4 mmol m<sup>-2</sup> d<sup>-1</sup>) were not significantly different between the CL and HI sites, suggesting that denitrification rates in the net area remained stable despite trawling activities if the NO<sub>3</sub><sup>-</sup> flux can be assumed to be a proxy for denitrification. Model results by De Borger et al. (2021) applicable to the net area predicted that increased oxygenation due to MBCF may double denitrification rates by enhancing NO<sub>3</sub><sup>-</sup> availability, possibly due to higher rates of coupled nitrification-denitrification. In contrast, intensive fishing activities may reduce NO<sub>3</sub><sup>-</sup> concentrations through increased microbial activity, excessive sediment mobilization, and net N loss from sediments (Hale et al., 2017). Ferguson et al., (2020) also reported that benthic trawling altered sediment redox profiles and reduced denitrification rates by 50%, based on N<sub>2</sub> flux data. The similarity of NO<sub>3</sub><sup>-</sup> fluxes between the CL and HI areas could thus be driven by a combination of lower denitrification rates in combination with NO<sub>3</sub><sup>-</sup> loss to the water column. This variability of the trawling impact on the direction of the benthic fixed N source/sink, in addition to the variety of methods used, highlights the confounding complications of site-specific environmental factors, such as benthic fauna, nutrient availability and oxygen penetration depth on the N cycle (Sciberras et al., 2016).

Our data nonetheless show that trawling leads to a net decline in the benthic fixed N source to the water column. The exact cause of the N loss (e.g. increased denitrification versus porewater NH<sub>4</sub><sup>+</sup> efflux to the water column) is difficult to constrain and would require detailed laboratory studies to gain further insight. Post-trawling porewater concentrations of PO<sub>4</sub><sup>3-</sup> were elevated immediately below the sediment surface (Fig. 6), and it follows that scraping off surface sediment by trawling mixes surface porewater PO<sub>4</sub><sup>3-</sup> into the lower water column, as previous findings have shown (Bradshaw et al., 2021; Duplisea et al., 2001; Sciberras et al., 2016). We currently lack data on bottom water chemistry immediately after trawling to track the missing PO<sub>4</sub><sup>3-</sup> after sediment resuspension. Since P cycling in sediments is to a large degree controlled by the redox cycling of Fe (Slomp et al., 1996), it remains to be shown whether the lower PO<sub>4</sub><sup>3-</sup> flux was instead caused by adsorption onto freshly formed iron (oxyhdr)oxides following resuspension (Almroth-Rosell et al., 2012). The observed reduction in porewater Fe<sup>2+</sup> concentrations after trawling could be evidence for this, although presently this is only speculation.

Despite the large number of in situ lander operations over the 16-day experimental period, the recovery time of the sediments with regards to fluxes is difficult to unequivocally reconcile for all chemical components (Fig. 8). DIC, TA and H<sub>4</sub>SiO<sub>4</sub> were consistently higher in the undisturbed control area, whereas for the other nutrients the HI fluxes seemed to converge to the CL fluxes soon after trawling. Yet, the temporal variability of the control fluxes makes it hard to define a robust control baseline, and this is a crucial point that should be noted for all published and planned fieldwork studies on this topic. The temporal decrease in bottom water O<sub>2</sub> is an additional complication since it will affect the rates of carbon cycling and associated redox processes. However, results from long-term (~weeks) ex situ core incubations in the oxygen minimum zone off NW Africa show that benthic fluxes only begin to be altered when dissolved O<sub>2</sub> falls below ~20  $\mu$ M and with a time lag of 1-2 days (Schroller-Lomnitz et al., 2019). Bottom water dissolved O<sub>2</sub> at Mecklenburger Bight was consistently above this threshold (Fig. 2).

# 4.2 MBCF and organic carbon remineralization

The organic carbon remineralization rate calculated from our benthic flux measurements (RPOC<sub>TOT</sub>) showed a decline following trawling activities. This reduction is further evident from the in situ DIC and TOU flux measurements. Understanding the impact of trawling on ocean carbon storage is key to predict anthropogenically-induced changes to the benthic carbon cycle, yet it remains a subject of considerable debate (Epstein et al., 2022; Zhang et al., 2024). Trawling-induced mechanical disturbance can increase the oxygen exposure of POC buried in sediments, thereby promoting aerobic remineralization and subsequently affecting atmospheric CO<sub>2</sub> uptake (Atwood et al., 2024; Sala et al., 2021). Although oxygen exposure is a major control on organic matter remineralization (Dauwe et al., 2001), other factors are also important. These include the age and composition of organic matter, associated biological communities such as microbes, particle grain size and mineralogy as well as

the local hydrodynamic conditions (Aller et al., 1996; Arndt et al., 2013; Arnosti and Holmer, 2003; Burdige, 2007; Hedges and Keil, 1995).

The surface layer of the sediments holds a high abundance of microbial and macrofaunal communities (Dauwe and Middelburg, 1998; Watling et al., 2001). Our porosity profiles (Fig. 5f) strongly implies that the upper 2 cm might have been scraped away by the ground rope, in agreement with other trawling experiments (Tiano et al., 2019; Bradshaw et al., 2021). The physical disturbance of the surface sediments along with the removal of fauna may decelerate biogeochemical processing and thereby reduce carbon remineralization rates (De Borger et al., 2021; Morys et al., 2021; Pusceddu et al., 2014; Tiano et al., 2019). Fishing experiments conducted in the North Sea reported a decline in benthic metabolic activity immediately after sediment disturbance (Tiano et al., 2019), whereas other model studies suggest little change (Rooze et al., 2024) or even an increase (e.g. van de Velde et al., 2018).

Sediment grain size plays a crucial role in POC remineralization, since fine-grained particles act as preservation 625 nuclei for organic carbon (Hedges and Keil, 1995). A sediment resuspension experiment conducted in the Kiel Bight (Baltic Sea) suggests that grain size controls the difference between remineralization rates under oxic and anoxic settings, with higher rates under oxic conditions for coarser sediments (Kalapurakkal et al., 2025). At Mecklenburger Bight, the sediments are predominantly fine-grained, which should help to protect against carbon remineralization by adsorption onto mineral surfaces. However, although the abundance of fine-grained sediment 630 plume was very prominent, we made no attempt to determine the fate of sediment resuspended in the sediments. Furthermore, Tiano et al. (2019) observed that while reductions in total organic carbon may be subtle, MBCF disproportionately affects the more reactive fractions of organic carbon. In certain depositional environments, fine-grained suspended sediments may be transported and redeposited in different locations, potentially increasing seabed POC content in downstream areas (Epstein et al., 2022; Palanques et al., 2014). Continuous scraping and 635 ploughing of the surface sediments can also lead to the enrichment of POC due to the upliftment of organic-rich deeper sediments (Palanques et al., 2014). Based on the evidence at hand, we suggest that the reduction in POC remineralization rates shown by our flux measurements can be attributed to the curtailment of biological respiration caused by the removal of the sediment surface.

# 4.3 Impact of benthic alkalinity losses on air-sea CO2 fluxes





Literature discussions on the impact of trawling on air-sea CO<sub>2</sub> exchange have focused on the oxidation of resuspended organic carbon as the principle agent of enhanced CO<sub>2</sub> release (Atwood et al., 2024; Hiddink et al., 2023; Sala et al., 2021). Yet, it has recently been shown that pyrite oxidation in the water column induced by sediment resuspension has a far higher potential to induce CO<sub>2</sub> emissions to the atmosphere and a weakening of the carbon shelf pump (Kalapurakkal et al., 2025). This is particularly significant because pyrite formation and burial are major contributors to alkalinity generation and play a crucial role in long-term carbon storage (Hu and Cai, 2011; Reithmaier et al., 2021). A study by van de Velde et al. (2025) further indicated that trawling and dredging reduce natural alkalinity production, thereby weakening the marine carbon sink. Their global estimate indicates that this loss of alkalinity could result in the release of approximately 4.9 Tg CO<sub>2</sub> per year from the ocean to the atmosphere. Our results add to this body of knowledge by showing that trawling also causes a large and immediate reduction in benthic DIC and TA emissions by roughly 50 % (by 9.3 and 6.8 mol m<sup>-2</sup> d<sup>-1</sup>, respectively), which then persists for at least two weeks. The observed reduction in TA and DIC in the present study may be attributed to decreased remineralization of POC in addition to the removal of the undersaturated layer were calcite dissolution is occurring. Once resuspended, it is likely that remineralization of organic carbon to DIC will continue to some extent, although the associated release of alkalinity will be limited to ammonium release. From a benthic perspective, our data demonstrate a pronounced temporal alteration of ecosystem functioning and structure. To the best of our knowledge, these are the first in situ measurements of TA and DIC fluxes in the net area disturbed by trawling.

Data on TA fluxes determined in situ using benthic chambers in the Baltic Sea are rare. Previously, using the BIGO landers, TA fluxes of 7 – 23 mmol m<sup>-2</sup> d<sup>-1</sup> were determined for the Fehmarn Belt and 8 – 30 mmol m<sup>-2</sup> d<sup>-1</sup> in fine-grained sediments of the Eastern Gotland Basin (Sommer et al., unpublished data). In the Gulf of Gdansk, diffusive carbonate alkalinity fluxes of 1 – 2 mmol m<sup>-2</sup> d<sup>-1</sup> have been reported (Łukawska-Matuszewska and Dwornik, 2025). High benthic TA release thus appears to be characteristic of muddy sediments in the Baltic Sea, which may partly be caused by benthic carbonate dissolution (Wallmann et al., 2022). In muddy Baltic Sea

sediments, carbonate dissolution is driven by mineral undersaturation in a thin subsurface layer arising from the acidity produced by aerobic respiration pathways (Dale et al., 2024). Carbonate dissolution results in the release of two moles of alkalinity per mole of DIC:

$$CaCO_3 + CO_2 + H_2O \rightarrow 2HCO_3^- + Ca^{2+}$$
 Eq. (7)


An estimate of carbonate dissolution rates (R<sub>CaDiss</sub>, mmol CaCO<sub>3</sub> m<sup>-2</sup> d<sup>-1</sup>) in the CL area can therefore be made as follows (e.g. Burdige, 2006; Hammond et al., 1996):

$$R_{CaDiss} = 0.5 \cdot (J_{TA} + J_{NO3} - J_{NH4} - 4 \times J_{PyB})$$
 Eq. (8)









TA and DIC emissions.

where  $J_{TA}$  (12.2 mmol m<sup>-2</sup> d<sup>-1</sup>) is the mean TA flux at the CL sites,  $J_{NO3}$  is the NO<sub>3</sub><sup>-</sup> influx into the sediment (0.38 mmol m<sup>-2</sup> d<sup>-1</sup>) and  $J_{NH4}$  is the ammonium efflux (2.2 mmol m<sup>-2</sup> d<sup>-1</sup>). The burial flux of pyrite  $J_{PyB}$  (as moles of TA) has been calculated to be 0.9 mmol m<sup>-2</sup> d<sup>-1</sup> using the model described below; similar to values from a global model analysis of coastal muds by van de Velde et al. (2025). Inserting the numbers gives  $R_{CaDiss} = 4.7$  mmol m<sup>-2</sup> d<sup>-1</sup>, or 9.4 mmol TA m<sup>-2</sup> d<sup>-1</sup>, which demonstrates that carbonate dissolution is a likely candidate for the bulk of the measured TA flux. This number is near-identical to a global model analysis of coastal muds by van de Velde et al. (2025).

Given widespread trawling activities in the region, it is pertinent to question the broader impact of a ramping down of benthic TA and DIC source by bottom fishing on CO<sub>2</sub> exchange with the atmosphere. A coupled benthicwater column model was used to estimate the impact. The model was previously used to examine the impact of sediment resuspension on air-sea CO<sub>2</sub> exchange in the western Baltic Sea (Kalapurakkal et al., 2025). The sediment compartment accounts for the rain rate of POC to the seafloor, degradation of POC coupled to pyrite production, pyrite oxidation caused by shallow trawling (upper 2 cm, i.e., resuspension by net scraping) and pyrite and POC burial, but did not consider carbonate dissolution. Also considered is the enhanced degradation of POC by exposure to oxygen following trawling although, as mentioned, this has only a minor impact on CO<sub>2</sub> fluxes compared to that induced by pyrite oxidation. The corresponding benthic DIC and TA fluxes are fed directly back to the well-mixed water column compartment in a fully coupled manner. In the water column, there is lateral exchange of TA and DIC, and a flux of CO<sub>2</sub> to or from the atmosphere depending on the strength of the benthic

Initial environmental parameters were set to the study site values (water depth, salinity, temperature, water column DIC and TA concentrations). The molar ratio between pyrite formation and POC oxidation was adjusted to match the local pyrite contents in CL sediment (0.47 wt.%). Following Kalapurakkal et al. (2025), we obtained the trawling frequency (2 yr<sup>-1</sup>) and disturbed areas (38%) from the ICES Vessel Monitoring by Satellite (VMS) database (for calculation see Amoroso et al., 2018). Trawling was assumed to expose the upper 2 cm to oxygen over a five-day period. Benthic DIC fluxes in the original model were exclusively due to POC degradation, whereas TA fluxes were driven by pyrite burial and oxidation. In this study, the model was modified in two ways; (i) to include baseline values of carbonate dissolution at the derived rates of 9.4 mmol TA m<sup>-2</sup> d<sup>-1</sup> and 4.7 mmol DIC m<sup>-2</sup> d<sup>-1</sup>, and (ii) during each trawling event, the benthic DIC and TA fluxes were reduced from the baseline values by 9.2 and 6.8 mmol m<sup>-2</sup> d<sup>-1</sup> in accordance with the flux data in Fig. 7, and for a period of 21 days. With this approach, we consider that carbon degradation and carbonate dissolution were diminished since most of the degradation and dissolution takes place in surface sediments that were partly removed by trawling. A description of the empirical constraints on trawling frequencies and depths can be found in Kalapurakkal et al. (2025). Our updates to the model are described in the Supplement.

The model results show that each trawling episode leads to pronounced losses of TA, as expected from previous results (Kalapurakkal et al., 2025). Sediments transition from a TA source (positive fluxes) to a large TA sink (negative fluxes) during the first trawl due to pyrite oxidation ("Standard run" red curve in Fig. 10a). Note that only a few percent of ambient pyrite pool need to be oxidized per trawl to account for the TA losses (Kalapurakkal et al., 2025). TA fluxes decrease for each subsequent trawling event due to a long-term impoverishment of pyrite

stocks (Kalapurakkal et al., 2025). The water column is a continuous source of CO<sub>2</sub> to the atmosphere at the study site, with large emission peaks consistent with the pulse of acidity produced by oxidation of resuspended pyrite ("Standard run" red curve in Fig. 10b).

In a second execution of the model, DIC and TA fluxes were not reduced by the values of 9.2 and 6.7 mol m<sup>-2</sup> d<sup>-1</sup> that we observed during trawling. A comparison of the results of this exercise ("No directly imposed TA reduction" black curves in Fig. 10) with the standard run shows that the observed reduction in TA and DIC fluxes by trawling has a very minor, almost negligible, impact on TA fluxes and  $CO_2$  emissions. Integrated over the 10 years, the total  $CO_2$  fluxes to the atmosphere in both model scenarios differ by < 1 %.






These initial results show that scraping of the surface sediment in the net area by sweep lines, foot ropes and fishing lines hanging between the otter boards has a demonstrable impact on enhancing atmospheric CO<sub>2</sub> emissions at the study site. However, this is almost entirely driven by pyrite oxidation, as shown previously for the Kiel Bight (Kalapurakkal et al., 2025). Lower measured surface pyrite contents in the HI versus CL areas tentatively fit with this finding. Changes in benthic DIC and alkalinity fluxes observed during trawling are related to the declined in POC degradation and carbonate dissolution caused by the removal of reactive surface sediments. These coupled processes have only a small net effect on atmospheric CO<sub>2</sub> since carbon degradation promotes CO<sub>2</sub> release whereas carbonate dissolution induces CO<sub>2</sub> uptake from the atmosphere. The observed decrease in benthic alkalinity fluxes by net scraping provides a very small additional weakening of the shelf carbon pump. Consequently, whilst the physical destruction of the sediment surface and enhanced pyrite oxidation are real causes for environmental concern, the additional and immediate reduction of the background alkalinity flux due to the physical removal of the surface sediment appears to be unimportant for the water column carbonate system. Important open questions remain for further investigation, including ultimate physico-biogeochemical reasons for the decline in TA and DIC fluxes observed after the trawling, the impact on aerobic versus anaerobic POC degradation, and a deeper understanding on the exact fate of resuspended sediment, including reduced iron phases.

**Figure 10:** Results from the box model. (a) Benthic TA fluxes, and (b) CO<sub>2</sub> emissions to the atmosphere (negative values). Positive TA fluxes represent benthic emissions and vice versa. The red curves show the standard model run using the observed decrease in TA and DIC fluxes following trawling, and the black curves show the fluxes

without these decreases. The vertical grey lines indicate annual trawling events. Note that the red and black curves are mostly superimposed. DIC fluxes are qualitatively similar to TA fluxes (not shown).

# 745 5. Conclusions







This field study demonstrates that MBCF has a profound and lasting impact on benthic remineralization processes. Seafloor imaging vividly illustrates the significant disruption to seafloor morphology, resulting in the resuspension of a sediment plume. Trawling is shown to cause substantial reductions in the fluxes of DIC, TA and nutrients in the net area, which do not fully recover to baseline levels by the end of the 16-day sampling period. The displacement of surface-active sediments directly impedes overall benthic biogeochemical processes, leading to diminished benthic fluxes and a marked reduction in the POC remineralization. Our data underscore that a reduction in biological respiration following the removal of the sediment surface is a key driver of the lower DIC and TA fluxes. A coupled benthic-water column model provides insight into the broader implications of benthic TA losses for CO<sub>2</sub> exchange. Model results suggest that while trawling-induced reductions in TA and DIC fluxes influence carbon cycling, their impact on atmospheric CO<sub>2</sub> sequestration remains minor compared to trawling-induced pyrite oxidation.

To fully understand the long-term consequences of MBCF, further research should investigate the fate of resuspended sediments, particularly their mineralogical composition and biogeochemical reactivity. However, the temporal and spatial variability of the fluxes within the relatively small control area complicates the characterization of an undisturbed baseline; a caveat that is widely relevant for field studies. Future site-specific studies must also consider local oceanographic conditions and high-trawling-intensity areas to comprehensively assess the cumulative effects of trawling on benthic ecosystems. Such research is critical for informing sustainable fisheries management and mitigating human-induced disturbances in marine environments.

# 765 Acknowledgements

This work received funding awarded to SS and AD by the German Federal Ministry of Education and Research (BMBF), Grant No. 03F0937F, Project MGF-Ostee, DAM pilot mission: Exclusion of mobile bottom-contact fishing in marine protected areas of the German EEZ of the North Sea and Baltic Sea (MGF North Sea and MGF Baltic Sea). We thank Bettina Domeyer, Regina Surberg, Anke Bleyer, Alexandra Subic, and Frauke Langenberg for assistance during the collection and analysis of sediment samples, and Matthias Türk and Asmus Petersen, and Alexandra Subic for gear preparation and deployments. We are also grateful to Gabriel Nolte, Jens Greinert and Martin Pieper for assistance with the XOFOS. The captain and crew of RV Alkor are warmly thanked for their professional and easy-going work environment on board. We thank Daniel Oesterwind for his help with the reviewers' comments on the otter trawl specifications. We greatly appreciate the insights and constructive comments provided by Sebastiaan van de Velde and Sarah Paradis. And finally, to Jack Middelburg for editorial handling of our manuscript.

**Competing interests:** The authors declare that they have no conflict of interest.

**Author Contributions:** S.S. and A.W.D. were responsible for planning and executing the study, as well as funding acquisition. S.S. conducted the BIGO and XOFOS deployments. P. L., H.T.K., and A.W.D. were involved in sediment sampling and analysis. A.W.D. conducted the modeling work. J.K. and S.B. carried out the sulfate reduction measurements, while F.S. performed the pyrite measurements. P. L., A.W.D., and S.S. prepared the manuscript with contributions from all authors.

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
