# Peer review of "Reduction of carbon, alkalinity and nutrient fluxes in the southern Baltic Sea caused by dragging of otter trawl nets across the seafloor"

_EGUsphere, 2025_

## Author Comment (AC1)

**Response to Reviewer 1**

This manuscript presents the results of a field investigation of the impact of dragging an otter trawl rope across the seafloor. This is a well-designed field study that makes great use of the unique in situ observational capacities of GEOMAR. It is overall well written and structured, and will be a valuable and nuanced addition to the current literature on the impact of mobile-bottom contact fishing (MBCF) on the marine carbon cycle. One comment on the writing is that I would urge the authors to consider tempering the tone of the text – hyperbolic words such as 'severe', 'dramatic', 'substantial' … are used often throughout the text, but they are not necessary given the nuanced nature of the data and main results of the study. While recent publications on MBCF impacts have tended to sensationalise, this is not aiding our understanding or helping to nuance the discussion around managing MBCF. A bit more scientific sobriety would be welcome in this specific field.

I also have a few concerns and suggestions on content that probably should be addressed before the manuscript will be ready for publication. I give some general comments here, with more detailed comments below.

I believe the authors will be able to address these, and I am looking forward for a revised version of this interesting manuscript.

**We thank Sebastiaan for constructive comments and suggestions on the manuscript. We have addressed them in detail in the revised version and in the response to the specific comments below.**

General comments:

1. The main observation is an overall reduction in benthic fluxes after disturbance, which the authors suggest is due to the erosion of the surface layer with more reactive organic matter and silicates. This is very likely correct, but throughout the discussion I feel this is sometimes forgotten (see my detailed comments on L544, L609, L694). The total impact of dragging ropes on the marine carbon cycle cannot be accounted for by only measuring before and after fluxes, as the fate of the eroded layer needs to also be considered. This should be better reflected throughout the MS.

Another effect that does not receive a lot of attention is the transient nature of the data. How much of the observed change in flux is transient due to the porewater build-up after erosion/mixing event – rather than reflecting actual changes of the biogeochemical pathways? For example, the way you calculate calculation RPOC_tot from the fluxes assumes steady-state, but if you flush out the top porewater, you will get a recovery phase where fluxes will be lower until the new steady-state is reached. So your estimation of RPOC after disturbance is an underestimation. Since the large variability in SR probably means the difference is not statistically significant can you confidently say there is a difference?

**We agree that the observed changes in fluxes may, in part, reflect transient dynamics resulting from porewater flushing following the erosion and mixing event. However, it is hard to say if the calculated RPOC is an underestimation in the sense of an artifact. RPOC is lower mainly because the reactive surface layer was physically removed during the disturbance, and as the system returns to a new steady state, the suspended material may also resettle, potentially contributing to flux recovery. Our data consistently show a significant difference in RPOC values (P-value = 0.037) between control and HI areas, supporting the interpretation of a real change in benthic carbon respiration. We also included in the conclusions that the fate of resuspended organic carbon in the water should be considered to appreciate the total impact of trawling.**

2. My final comment relates to the model usage and claim of pyrite oxidation. While I think this is indeed a factor that needs to be consider, I don't see any new evidence in the manuscript that this occurs. The modelling part of the MS, which is used to claim that pyrite oxidation is important, present essentially the same model runs (with minor variations) as previously done by (Kalapurakkal et al. 2025), and thus do not validate the conclusions of the earlier manuscript, nor do they bring much new to the table, since you are essentially getting the same results. At the very least, I would have expected the field data to be used to validate the model runs, but this is also not the case.

I would suggest that the authors reconsider the added value of the model simulations in their current form (see also my detailed comment below), and also ask them to consider how these results are presented, as the sentence in the abstract at L34 and L669 give the impression that this study is an independent validation of the earlier model results of (Kalapurakkal et al. 2025), which it is not.

**Following Kalapurakkal et al. (2025), we obtained trawling frequency (2 yr-1) and disturbed areas (38%) from the ICES Vessel Monitoring by Satellite (VMS) database. In that manuscript, aerobic pyrite oxidation was clearly shown to be occurring, and given the similar nature of the sediments between Kalapurakkal et al. (2025) and our present study sites we believe it is a solid assumption to assume that resuspended pyrite (which we measured) would also be oxidized on short timescales. We cannot use the field data to validate the model runs since, as explained in the paper, the geochemical transformations of resuspended water column were not tracked, and this should be done in future studies. The goal of the modeling exercise was to compare the impact of lower TA release following resuspension of the surface layer on air-sea CO2 fluxes. The model indeed shows that this is negligible compared to pyrite oxidation, and this is an important result that the scientific community should be aware of to understand the broader impacts of trawling on carbon and alkalinity budgets. We have modified the sentence on L669 to make clear that model result is theoretical and requires further validation.**

Detailed comments:

- Throughout: 'bottom trawling' – is supposedly colloquial, and a more accurate term is mobile bottom-contact fishing. Not so much for the experiment in this paper, as you are looking at otter trawl specifically, but for the more broad scope papers in the introduction.

  **Thank you for the suggestion. While we acknowledge that "mobile bottom-contact fishing" (MBCF) is the more precise term, "bottom trawling" remains widely used in the scientific literature. Therefore, we have retained "bottom trawling" throughout the manuscript for consistency and readability. However, we have now clarified in the Introduction (line 38) that "bottom trawling" refers to mobile bottom-contact fishing (MBCF).**

- Title: might be more appropriate to name that it was an otter trawl rope

  **We agree with your suggestion and modified the title as "Severe reduction of carbon, alkalinity and nutrient fluxes in the southern Baltic Sea caused by dragging of otter trawl nets across the seafloor".**

- L30: 'supresses benthic mineralization' – the reason that happens is because a lot of reactive POC is removed – so it does not so much supress it than displace it?

  **Thank you for pointing that out. Now we have modified the sentence to "Additionally, observed variations in organic carbon remineralization rates suggest that bottom trawling alters benthic respiration by disrupting key biogeochemical processes."**

- L50ff – might be worth to discuss the results of (Porz et al. 2024; Zhang et al. 2024) as well in the light of the carbon sequestration debate

  **We appreciate your suggestion and have revised the introduction accordingly. We have now included the following sentence: "In particular, trawling in shelf seas has been shown to reduce POC by 29% (Porz et al., 2024), with long-term losses equivalent to emissions of 3.67 Mg $CO_2$ km$^{-2}$ yr$^{-1}$, assuming complete mineralization of the disturbed POC (Zhang et al., 2024)."**

- L82ff – I would include the papers that show/discuss the importance for pyrite formation as an alkalinity source (Hu and Cai 2011; Reithmaier et al. 2021). And it might also be interesting to bring in our recent global estimate of the chronic impact of repeated trawling (van de Velde et al. 2025) – especially since there is a first-order estimate of alkalinity loss for the area in which you did the experiment (also see the SI for a long-term reduction in TA flux due to chronic trawling). There is also an estimate of how import each individual process is for shelf sediment alkalinity generation.

Thank you for bringing these papers to our attention. We have incorporated the above citation and revised the introduction section accordingly. The modified sentence now reads: "Several processes control alkalinity generation in sediments and fluxes to the water column, such as mineral formation and dissolution, denitrification, pyrite formation and burial, and reverse weathering (Hu and Cai, 2011; Krumins et al., 2013; Middelburg et al., 2020). Among these, alkalinity production associated with pyrite formation likely constitutes a significant blue carbon sink (Hu and Cai, 2011; Reithmaier et al., 2021). Experimental and modeling studies have shown that trawling-induced resuspension reduces the capacity of the Baltic Sea to remove atmospheric $CO_2$ by decreasing alkalinity, mainly through the oxidation of pyrite (Kalapurakkal et al., 2025). More broadly, bottom trawling and dredging activities are estimated to reduce alkalinity generation, thereby weakening the marine carbon sink by approximately 2–8 Tg $CO_2$ per year, through their impact on both organic and inorganic carbon cycling (van de Velde et al., 2025)."**

- L223: so you had the instruments (nutrient analyzer, alkalinity titrator, etc.) on board the ship?

  **Yes.**

- L288: unclear, is this 1 to 50 or 150 or?

  **It is 1 to 50 mL. The text has also been revised to avoid confusion.**

- Section 2.8 – Curious, how does this compare to the DIC flux?

  You can also do a similar exercise by including DIC and TA fluxes and making a similar mass budget, this time including carbonate dissolution and pyrite/FeS burial

  **Thank you for the comment. This calculation has already been carried out in Section 4.3 using Equation 8.**

- Figure 4: maybe say 'sediment cast', which makes it easier to directly understand the figure

  **We have modified the figure and changed to 'sediment cast'.**

- L432: but higher up you assume a different oxidation state for your organic carbon? Why not be consistent?

  **Thank you for pointing that out. We have now accounted for the zero oxidation state of carbon in our RPOC calculations and have updated the corresponding text and figure accordingly.**

- L454: lower near the surface, as they become higher at depth in the higher impact areas?

  **Yes, porewater concentrations (TA, $NH_4^+$, and $H_4SiO_4$) are lower near the surface and increase with depth.**

- L486: why not include this in this manuscript?

  **That work is an ongoing collaborative effort with other research groups and is not intended to be included in this manuscript.**

- L514: why surprisingly? You just describe yourself that your site is at the threshold where bottom-water O2 is controlling the O2 flux – so removing POC should not affect the O2 flux.

  **Thank you for pointing this out. We agree with your observation. The term "surprisingly" has been removed to better reflect the expected outcome based on the oxygen dynamics at the site, as discussed in the preceding text.**

- L518: could it also have to do with sediment type? A sandy sediment might be more prone to porewater flushing due to the disturbance, where a muddy sediment would be less. I can then see how muddy sediments would show higher fluxes right after recovery if the porewater is mixed rather than flushed out.

  **Thank you for this comment. We agree that sediment texture influences how porewater responds to physical disturbance. Both our study and that of Morys et al. (2021) were conducted in muddy sediments, which are less prone to flushing and more likely to exhibit mixing after disturbance. However, despite this similarity, our results contrast with theirs. In our case, the erosion of the reactive surface layer appears to have led to a net decrease in benthic fluxes, rather than a transient increase due to porewater mixing. It is possible that an immediate measurement of fluxes following the disturbance could have captured a short-term enrichment due to mixing, but our sampling likely captured a later phase when the impact of surface layer removal became dominant.**

- L533ff: are the studies you mentioned not directly determining the denitrification rates through modelling, isotope pairing, or N2/Ar fluxes? Whereas you are comparing it to the NO3 flux alone – which is not the same?

  **Yes, this is correct. We have now included this caveat in the text.**

- L544: I don't think I agree with that statement – if the loss of fluxes is due to the erosion and removal of the reactive surface layer, you need to also account for the fate of that surface layer before you can make claims about the impact. If the POC

gets remineralized in the water column, you still produce the nutrients, so you don't affect the productivity.

**We agree that it is not possible to conclude that the reduction in nutrients due to trawling directly affects water column productivity without understanding the fate of the suspended sediment layer. In response, we have removed these sentences.**

- L575: and probably most importantly: the nature of the organic matter itself (age, origin) – which to a large extent will determine its sensitivity to environmental conditions.

  **We agree with your comment and have revised the text accordingly, including supporting citations. The sentence now reads as follows: "These include the age and composition of organic matter, associated biological communities such as microbes, particle grain size and mineralogy as well as, the local hydrodynamic conditions (Aller et al., 1996; Arndt et al., 2013; Arnosti and Holmer, 2003; Burdige, 2007; Hedges and Keil, 1995)".**

- L590: in what way? Coarser grain size = bigger difference?

  **Yes, now clarified.**

- L590: Our study from the anoxic Baltic Sea suggests that low mineral protection (high OM concentrations and low sediment accumulation) leads to high mineralization rates, even under anoxic conditions (van de Velde et al. 2023; Placitu et al. 2025). This indicates that the lack of mineral protection leads to no difference in oxic versus anoxic conditions – rather the inverse of the interpretation of the results of Kalapurukkal.

  Could it be that the results of Kalapurukkal actually show the effect of desorption and the age of the organic matter? Fine-grained sediments would protect OM from mineralization, meaning more reactive fractions remain. When incubated in suspension, desorption occurs in both oxic and anoxic conditions – and since more reactive OM fractions show little difference in mineralization rate under oxic or anoxic conditions (see the earlier work of, e.g., (Kristensen et al. 1995)), you observe                                      little                                      difference.

  With coarse-grained sediments, there is little mineral protection, and the more reactive fractions have quickly reacted away. When you then incubate the sediment in suspension, the less reactive OM fractions show differences in mineralisation in oxic versus anoxic conditions.

  This would lead to a slightly different mechanistical interpretation of the results and would reconcile it with our findings. It is not the grain size that controls the response of mineralization in oxic versus anoxic conditions, but grain size that controls which

OM fractions are retained in the sediment – and this eventually is reflected in the resuspension experiments.

**Thank you for these insights. Our intention was simply to highlight that the fine-grained nature of our sediments may promote organic carbon preservation through adsorption onto mineral surfaces. This manuscript is not the place to discuss the fate of resuspended organic matter since we did not attempt to collect it. However, these are important considerations for our ongoing resuspension experiments.**

- L606: also considering including our recent global estimate (van de Velde et al. 2025), and papers that discuss that sedimentary pyrite burial is an important source of alkalinity (Hu and Cai 2011; Reithmaier et al. 2021)

  **We have included the citations in the revised text.**

- L609: but what about the fate of the resuspended material?

  **We have now added: "Once resuspended, it is likely that remineralization of organic carbon to DIC will continue to some extent, although the associated release of alkalinity will be limited to ammonium release."**

- L611: 'dramatic' – a bit over-the-top, since you show a temporary reduction in fluxes, how does that say anything about ecosystem functioning or structure?

  **Agree. Now modified to "pronounced temporal…"**

- L612: 'to the best of our knowledge' – remove sentence, this does not add to the manuscript

  **We prefer to keep it, in case there are other studies out there that we are unaware of.**

- L634: this – interestingly – is exactly the number that comes out of our global modelling exercise (van de Velde et al. 2025), and also close to the numbers of (Krumins et al. 2013). Would also be worth referencing some earlier work on carbonate dissolution in muddy sediments (Aller 1982; Green and Aller 2001)

  **We have now updated the references and comparison with van de Velde et al. (2025). Note that we corrected a typo in Eq. 8 whereby the new calcite dissolution rate is 9.4 mmol m$^{-2}$ d$^{-1}$ compared to the previously reported value of 9 mmol m$^{-2}$ d$^{-1}$.**

- L650: Is there no data from trawling intensity/disturbed area for the region you are studying? Would probably be worth checking (e.g., (Amoroso et al. 2018) or

(Eigaard et al. 2017; Rickwood et al. 2025)) to do more realistic simulations or some sensitivity tests.

**Yes. Following Kalapurakkal et al. (2025), we obtained the trawling frequency (2 yr⁻¹) and the percentage of disturbed areas (38%) from the ICES Vessel Monitoring by Satellite (VMS) database, calculated using the equation provided by Amoroso et al. (2018).**

- L652: So you assume no impact on carbonate dissolution? Why? It is your biggest source, and if you reduce organic matter mineralization in the sediment, you will reduce porewater acidification and this carbonate dissolution rates? Note that our model study did not find any impact, but we did not erode the top layer, but let it settle after resuspension.

  **Yes, it is included at the rates we derived from the mass balance (Eq. 8).**

- L657: The way this model runs are explained are a bit confusing to me – 'no impact' is still impact, right? You induce mixing and get pyrite reoxidation? So the only difference with the 'standard run' is that you force the benthic fluxes – but are those not a consequence of the disturbance? Should you then not use your observed fluxes to validate the model, rather than run the model to upscale something which is actually not based on your observations?

  **Yes, this was misleading. We have now renamed the "no impact" run to "No directly imposed TA reduction". We do not fully understand the second part of the comment since the model is in fact based on the observations.**

- L671: the paper from Kalapurakkal is a bottle incubation experiment, so I would not really say this paper shows that it happens in reality. The paper suggests that pyrite oxidation is more important that the organic matter impact – and this study seems to be a validation – or at least should be, because at the moment it seems you are using their paper to claim pyrite oxidation happens, without actually showing any data that backs that claim.

  **That study was indeed a bottle experiment, yet we argue that aerobic pyrite oxidation does happen in reality, either in the water column or after settling to the sediment surface. We cannot definitively state that pyrite oxidation is occurring at our study site since we did not perform experiments to verify this; something that should be done in future. We simply applied the empirical model of Kalapurakkal to compare the potential impact of pyrite oxidation with the loss of benthic TA fluxes in $CO_2$ exchange.**

- L673: but what drives that reduction in alkalinity fluxes? Pyrite oxidation should be reflected in these fluxes as well.

**The loss of alkalinity is probably a combination of lower rates of anaerobic POC degradation, in addition to the removal of the undersaturated layer where calcite dissolution is occurring. We have now included this in the text.**

- L694: only because you do not account for the fate of the resuspended material

  **That is partially correct. The fate of suspended sediments and the associated organic matter remains uncertain, particularly regarding the extent to which they undergo remineralization or eventually settle. While our current dataset clearly demonstrates diminished fluxes and reduced POC remineralization, a key limitation of our study is the unknown fate of the resuspended sediment plume. This limitation is explicitly acknowledged in both the discussion and conclusion sections.**

- L699: but you do not present evidence for the oxidation of pyrite ?

  **No, for this reason, we only "suggest" that it takes place, given the predictive capacity of the model (see previous comments)**

---

## Author Comment (AC2)

**Response to reviewer 2**

In this study, Linsy and co-authors perform a very thorough experimental study to assess the biogeochemical effects of sediment disturbance by bottom trawling in the Baltic Sea, where the authors provide valuable new evidence highlighting the complexity of such disturbance on biogeochemical pathways.

Notably, this is the first study that assesses the impacts of bottom trawling on total alkalinity fluxes from an experimental perspective rather than relying solely on modelling. It also stands out for addressing the effects of demersal fisheries from different perspectives, providing a deeper understanding of the biogeochemical consequences of demersal fisheries. By performing obtaining different sampling types (CTDs, sediment cores, landers) and analyzing a wide range of parameters, the authors provide a deeper understanding of the different biogeochemical processes affected by this sediment disturbance, while also recognizing the limitations of their approach.

I thoroughly enjoyed reading this in-depth study, and commend the authors for the work behind this. While the manuscript is both timely and highly relevant, I do have a number of comments and questions – particularly regarding the experimental design, data analysis, and its interpretation – which I hope will help strengthen the overall clarity and impact of this study.

**We thank you, Sarah, for taking the time to thoroughly review our manuscript. We greatly appreciate your constructive feedback, which has helped us improve the clarity and quality of our work. Please find our detailed responses to each of your comments below.**

Main comments:

1. While the description of the methodology is very detailed and could serve as a guideline for future studies that aim to better understand the biogeochemical impacts of demersal fisheries given its broad scope, it is not clear to me what kind of experimental design this study is following. I had to re-read the methods to properly identify if it was a Control-Impact experimental design, or a Before-After Control-Impact experimental design (sample all sites before the disturbance to account for temporal and site variability). I initially thought it was a Control-Impact experimental design, but when looking more closely at Table S1, I noticed that the authors also sampled the impact site before (July 19) the experimental trawl (July 20), sort of making it a BACI experimental design (only sampled the impact site before disturbance). The authors should be clearer about this experimental study design.

   Being a BACI experimental design, the authors should perform statistical analyses not only comparing Control-Impact, but also prior to the disturbance. In addition, the continuous sampling 16 days after the disturbance to assess the recovery is done in comparison to the control site, but it should also be done in comparison to pre-disturbance.

This also raises concerns with the statistical analysis used. From my understanding, the authors combine the sediment profiles of the cores (Figs. 5-6), or the fluxes (Figs. 7) in the impact and control sites. However, since the data was collected in different periods with respect to the disturbance, which the authors plot in Fig. 8 to assess the recovery after disturbance, then combining the data assumes that the temporal variation is not relevant. A quick look at the raw data in Table S1 doesn't show differences in the fluxes of the MUC in the impact site prior to the disturbance in comparison to the fluxes after the disturbance.

The following comments assume that the data processing was correctly done, but this should definitely be looked into.

**Thank you for bringing this to our attention. Our study follows a control-impact experimental design. The two cores collected before the trawling event in the HI area are considered control samples, as now indicated in both the text and the table. However, performing statistical analysis on porewater fluxes based on only two data points is not advisable (BACI approach). Additionally, we did not have any BIGO deployments prior to the trawling experiment at these specific sites. We have now updated Figures 5 and 6 to include these samples within the control site category. As a result, the mean fluxes and porewater concentrations have changed slightly, but not substantially, as to alter the conclusions of the study.**

**Regards Fig. 8, the temporal changes are indeed included in the mean values. These are, of course, important. Yet, since we are comparing the CL and HI areas, both of which are undergoing short-term temporal changes, our results nonetheless show that the difference due to trawling is larger than those caused by natural temporal variability in the CL areas.**

2. In relation to the statistical analyses (section 2.9), the authors consider that a p-value $< 0.05$ is indicative of statistically significant variability. After resolving the issue of my previous comment, I suggest the authors be more precise about the statistical significance of their results, and provide more detail about their statistical significance. For instance, the authors could add an asterisk in the figures to denote statistically significant differences with different confidence values, such as * for $p < 0.05$, ** for $p < 0.01$, and *** for $p < 0.05$ (or similar notation). I am surprised that there are statistically significant differences in the POC content of the surface sediment layers (Fig. 5a) considering that the limits of the error bars are touching (1 standard deviation, equivalent to 66 % of the variation of both samples). I have the similar doubts with the boxplots of TOU, TA, ammonium and phosphate of Fig. 7, since the upper and lower quartiles of the control and impact sites cross each other.

**Thank you for this useful comment. We have now clearly stated the significance thresholds in the Methods section (*$p < 0.05$, **$p < 0.01$, *** $p < 0.001$) and consistently indicated the significance of p-values in the Results section. The**

**POC values show a statistically significant difference in the surface (0–1.5 cm) section, with p-values of 0.046 and 0.036, respectively. The total fluxes of TA, ammonium, and phosphate also show significant differences between the control and HI treatments. TOU does not show a statistically significant difference. Regarding diffusive fluxes, TA is not significantly different, whereas ammonium (P value - 0.02) shows a significant difference even though the boxes overlap slightly.**

3. I am also missing some more background information of the study area, more specifically in relation to the fishing history. As pointed out in a data compilation of studies assessing the biogeochemical impacts of demersal fisheries, Paradis et al. (2024) identified that the control site in the majority of studies have been historically fished and were not being trawled during the study due to a seasonal closure or the recent establishment of a trawl ban. I am aware that the Baltic Sea has been extensively impacted one way or another (HELCOM, 2018; Bradshaw et al., 2024; Díaz-Mendoza et al., 2025), so what is the fishing history and current condition of the study area?

   In addition, how does the experimental fishing conducted in this study compare to the bottom trawling activities that usually take place in the Baltic Sea in terms of gear type, fishing intensity, fishing season? This is especially important to clarify and apply for the last modelling exercise (see comment 9). What is the distance between trawl tracks (red lines in Fig. 1), what is the width of the trawl nets and sweeplines? The authors have a schematic diagram of the gear type used in Figure 4, but this one is too small to annotate these elements (e..g, width between otter doors). This would be especially beneficial considering that the authors target the wide area between the otter boards (it would give additional perspective of why they target this area and not the furrows caused by the heavy otter doors).

   **The study area lies within the 3-nautical-mile zone, where fishing activities require special permission. Prior to the experiment, a detailed bathymetric survey was conducted, revealing no visible fishing tracks or seabed disturbances. However, according to HELCOM 2021 data, the broader Mecklenburger Bight experiences a maximum trawling intensity of 2 yr⁻¹ with 38% of the seafloor affected. We agree that the control area is defined as "control" only with regard to our experiment and is not intended to infer that the control area has never been trawled.**
   **It should be noted that this work represents a case study, and direct comparison with the entire Baltic Sea fishery is challenging, as different types of fishing gears are employed across the region. For this study, we used a standardized bottom trawl commonly applied in Baltic demersal surveys (TV-3#520 × 80 mesh size). The gear was fitted with ThyborØn Type 2 Standard trawl doors, each with a surface area of 1.78 m². The distance between the otter boards on either side of the trawl net was approximately 60 m. The length of the sweep line was 75 m. Additional technical specifications of the trawl configuration are provided in the ICES Baltic International Trawl Surveys (BITS) Manual. We have now updated**

**the manuscript to include this information and added a schematic diagram of the otter trawl. The sampling position was determined based on the trawl track recorded by the multibeam bathymetry.**

4. The contact of demersal fishing gear with the seafloor has several effects: it can resuspend sediment and hence erode the seafloor, create furrows associated to this sediment resuspension and erosion and adjacent sediment piles, and/or mix the sediment. The magnitude of each of these impacts is difficult to quantify, and the effects on sediment biogeochemistry will differ depending on these processes.

The authors observe a combination of these processes in this study: defined furrows, sediment piles next to the trawl tracks, visible scrapes, and a sediment plume implying sediment resuspension (lines 392-399; Fig. 4). What were the sizes (width and depth) of each of these features?

**The depth of the furrows ranged between 10–12 cm, and the width of a single trawl mark was approximately 85 cm. The precise physical impact of this disturbance is currently being prepared in a forthcoming manuscript.**

Later on, the authors conclude that bottom trawling has removed the upper 2 cm of sediment since there are statistically significant differences of POC in the impact and control site in these surficial sections (lines 414-416, Fig. 5). However, this reduction could also be due to remineralization, or mixing of the high OC in surficial 0-1 cm with the lower OC in deeper sediment sections (affecting the shape of the POC profile). Morys et al. (2021) observe that there is an upward 2.5 cm shift in the profiles of Chl-a, OM, and water content in the impact site (IN) with respect to the control site (OUT), which they attribute to erosion of these 2.5 cm. In this study, you also determined the water content and porosity of the sediment cores. This metric could be used to determine mixing (constant porosity in mixed layers) as well as erosion (removal of the less-consolidated surface sediment as seen by Morys et al. (2021)).The different physical effect of bottom trawling (mixing and erosion) and its biogeochemical effects should be discussed in more detail. For instance, the authors relate the lower flux of nutrients in the impact site due to erosion, but it could also be caused by mixing, which would accelerate the diffusion of porewaters to the overlying water.

**Thank you for your insightful comment. In line with the observations reported by Morys et al. (2021), we observed an upward shift of approximately 2 cm in the porosity profile at the HI site, strongly indicating erosion of the upper sediment layer. We have now incorporated this into Figure 5 of the revised manuscript to support our interpretation.**

**Although bottom trawling can also lead to mixing of surface sediments, our data more strongly support erosion as the dominant process. While some degree of mixing may occur, it does not appear to be the primary factor influencing nutrient fluxes in this study. If mixing were the main mechanism, we would expect an increase in total solute fluxes measured by the lander, as mixing enhances the release of porewater solutes into the overlying water. However,**

the observed reduction in total fluxes, particularly at the HI site, is more consistent with the removal of surface sediments rather than enhanced mixing. The BIGO measurements, which capture both diffusive and non-diffusive fluxes, further support our interpretation by providing a more comprehensive assessment of sediment–water exchange. Additionally, Figure 4 clearly shows the net  area (area affected by back strop, sweep lines, chains and bridles as well as by the foot rope and the fishing line), including the lines caused by the back strop as well as a thin layer of brown phytodetritus and underlying dark grey anoxic sediments. This suggests that the upper sediment layer has been removed. Further details on this observation will be provided in the aforementioned forthcoming manuscript.

5. In this study, the authors identify that sediment disturbance has minimal effects on TOU, in comparison to lowered $O_2$ consumption rates in other studies (Tiano et al., 2019; Bradshaw et al., 2024), implying that it is the first time this has been observed. However, as portrayed in a recent compilation of biogeochemical studies of the impacts of demersal fisheries (Paradis et al., 2024), there are several other studies that have shown that there is a minimal effect of demersal fisheries in oxygen consumption (Warnken et al., 2003; Polymenakou et al., 2005; Trimmer et al., 2005; Goldberg et al., 2014; Meseck et al., 2014). These studies were done in continental margins with contrasting dissolved oxygen concentrations, so the relationship between TOU and BW oxygen observed in this study are not necessarily applicable to those other studies.

   **Thank you for making us aware of your manuscript. The effect of trawling on TOU appears minimal in our study, consistent with findings from previous research. While we are not claiming this as a novel observation, we have now added a sentence in the revised text acknowledging that similar results have been reported in earlier studies, and we have cited the relevant literature accordingly.**

6. When discussing the mechanisms affecting the fluxes of nutrients, the authors discuss that phosphate could be released to the water column after disturbance, leading to a lower flux, but this is not the case for nitrate fluxes, since the fluxes of this latter nutrient did not vary between the control and impact sites (Fig. 7). If phosphate is released to the water column, nitrate should have been released as well (Breimann et al., 2022). Maybe the lack of significant difference of nitrate flux between the control and impact sites could be due to the counterbalance of nutrient release (as suggested for a decrease in phosphate flux) and decreased denitrification rates as reported in other studies (Ferguson et al., 2020).

   **Good suggestion that we have now included in the text.**

7. The authors find that in the impact sites, fluxes of DIC and TA decrease. They then mention that the biogeochemical explanation for this decrease is unclear (lines 610-612). Isn't it simply because there is a decrease in the RPOC? (Fig. 7h).

Also, the fluxes of TA in this study are within the range of TA fluxes in other regions (see lines 615-616). Hence, is there really an impact of bottom trawling in terms of TA flux?

**Yes, the reduction in TA and DIC may be attributed to a decrease in the rate of POC remineralization and calcite dissolution. Accordingly, we have removed the sentence stating that "the biogeochemical explanation for this decrease is unclear." The revised text now reads: "The observed reduction in TA and DIC in the present study may be attributed to decreased remineralization of POC and calcite dissolution (Fig. 7)." Although TA values in the present study are comparable to those reported from other regions, this does not imply that bottom trawling has no impact on TA fluxes. When compared to the control site, a clear reduction is evident.**

8. This is the first study that looks at the effects of bottom trawling in alkalinity fluxes. Van de Velde et al. (2025) performed a global modelling study of the effects of sediment disturbance on benthic alkalinity fluxes, and the causes behind it (carbonate dissolution, sulfate reduction and pyrite burial, denitrification). How do your observed results compare to those seen in that study? In that modelling study, they observe that the majority of the alkalinity reduction is due to changes in sulfate reduction and pyrite burial, but in your study, there are no statistically significant changes in sulfate reduction nor pyrite content. However, you mention that "sediments transition from a TA source to a large TA sink during the first trawl due to pyrite oxidation" (lines 658-659). I also don't agree with this sentence, since the sediments do not transition to a TA sink. Their TA flux is simply reduced in comparison to the control sites, so they are simply a "less strong TA source", to put it one way. What about carbonate dissolution? You calculate the carbonate dissolution rates in the control site, but what about the impact site? What about denitrification? See earlier comment 6.

**Our results for calcite dissolution are near-identical to the global model analysis of coastal muds by van de Velde et al. (2025), which is now included in the paper. Pyrite burial in our model equaled 1.5 mmol m$^{-2}$ d$^{-1}$; again, very similar to the global study (note that we corrected a typo in Eq. 8 whereby the new calcite dissolution rate is 9.4 mmol m$^{-2}$ d$^{-1}$ compared to the previously reported value of 9 mmol m$^{-2}$ d$^{-1}$). We believe it is correct to say that with trawling the sediment transitions to a TA sink because of the impact of proton release due to pyrite oxidation. The way the model is configured, pyrite oxidation represents a negative alkalinity source to the water column. We do not calculate carbonate dissolution for the HI site since here we are interested in the reduction in TA and DIC following trawling, which we ascribed to a reduction in POC degradation and carbonate dissolution (now mentioned). Denitrification (i.e. NO3 flux) is included in the mass balance (Eq. 8) to calculate carbonate dissolution.**

9. Regarding the seafloor-water-air box model of the impacts of DIC and TA fluxes, I don't understand the point of doing a trawling disturbance with and without changes

in DIC and TA fluxes. The experimental data show a reduction of DIC and TA fluxes, so why make a scenario called "No impact" but still with a trawling disturbance event?

**This was misleading. We have now renamed the "no impact" run to "No directly imposed TA reduction".**

The experimental approach was performed in summer, and the parameters observed during that season were applied in the box model for several years. Wouldn't the conditions change over time? Wouldn't the "baseline" $CO_2$ and TA fluxes also change seasonally?

**Most likely yes, but unfortunately, we only have data from one time point.**

Finally, is this fishing intensity (once per year) representative for this study area?

**Following Kalapurakkal et al. (2025), we have now obtained trawling frequency (2 yr-1) and disturbed areas (38%) from the ICES Vessel Monitoring by Satellite (VMS) database.**

10. Regarding future work, the authors modestly acknowledge in different points of the manuscript that they are not capable of discerning the causes behind the trends they see, and that they would need more information. What kind of information would they need to properly understand the causes behind the trends observed? For instance, the authors discuss why they don't see changes in nitrate flux in comparison to other studies, or decreases in phosphate, DIC and TA fluxes, and mention that they would need more information.

   The authors also acknowledge the need to study the fate of resuspended sediment to get a broader understanding of the biogeochemical consequences of sediment disturbance. Another aspect that should be studied is the effect of repetitive bottom trawling activities. Depending on the region, fishing grounds are disrupted almost on a daily basis, which would limit the capacity of deploying landers to properly calculate porewater fluxes, especially since the authors mention that porewater fluxes could be flawed (lines 489-500). This adds to the complexity of studying the biogeochemical impacts of demersal fisheries.

   **These are good points. More data for this kind of study is always welcome. Carefully controlled laboratory incubations to investigate Fe and P cycling in the resuspended layer is one potential research avenue that we are considering. We agree that repetitive trawling makes it problematic to properly assess the trawling impact on benthic fluxes. Recently, MPAs have come into force in the Baltic Sea, and continuous monitoring is now needed to compare future undisturbed fluxes with those in the same area that have been previously measured in our project. However, the temporal and spatial variability of the**

**fluxes within the relatively small control area complicates the characterization of an undisturbed baseline. We have now added this caveat to the conclusions.**

Minor comments:

- Line 22. The authors refer to studying the impacts on the "benthic ecosystem", which includes benthic communities (e.g., meiofauna). However, they did not study the benthic community. I would suggest to replace "benthic ecosystem" to "benthic biogeochemical pathways" or something similar.

  **We have replaced the benthic ecosystem with benthic biogeochemical pathways.**

- Line 28. Define which nutrients, since not all nutrients showed a decrease in flux.

  **Thank you for spotting this. Now we have mentioned that nutrients ($PO_4^{3-}$, $NH_4^+$, and $H_4SiO_4$) showed a reduction.**

- Line 29. Change "variations" to "decreases"

  **We have modified the text.**

- Line 32. Convoluted sentence. Suggest to modify to "[…] had not returned to baseline levels by the conclusion of the 16-day observation period, indicating prolonged effects of the disturbance, although natural temporal variations may have an influence." Or something similar.

  **We have revised the text as suggested.**

- Fig. 1. Add location of CTD deployment(s).

  **We have modified the figure to include the CTDs**.

- Lines 145-155. The authors extract porewater using two different approaches: centrifuge and Rhizon samplers. How do the results vary between both methods?

  **We employed two different methods due to a centrifuge malfunction that occurred midway through the cruise, and therefore could not compare the two methods. To minimize variability between the methods, instead of inserting Rhizons directly into the intact sediment cores, which is known to have artifacts due to the extraction of porewater from adjacent sediment layers, we first sliced the cores as usual and transferred the sediment into centrifuge tubes before inserting the Rhizons. While we cannot definitively quantify how this procedural difference may have influenced the results, we assume that its impact is minor.**

- Line 221. Were these geochemical analyses performed on the same sample (the same 1 cm interval of the same core)? I'm asking this because some samples were

treated (e.g., with ascorbic acid) and some were left untreated. Hence, aliquots would have had to be subsampled for each analysis, and I'm surprised you would have gotten sufficient volume for all of these analyses. Please clarify.

**Yes, all analyses were performed on the same set of samples. Approximately 10 mL of porewater was extracted from each sample and used for all measurements.**

- Lines 325-336. The diffusion fluxes were obtained from fitting the porewater data using the FindFit function in Mathematica software. As I'm not familiar with this software, and other readers may not be either, is this fitting done weighing the uncertainties of the measurement? Does this fitting give you a measure of uncertainty within a specific confidence interval? Was it error-propagated?

  **Thank you for this comment. The FindFit function in Mathematica was used solely to obtain a best-fit line through the porewater concentration data in the upper sediment layer. This approach allowed us to calculate a more representative concentration gradient for estimating diffusive fluxes, rather than relying on a simple linear gradient between the 0–1 cm depth interval. The fitting was not weighted by measurement uncertainties, and it does not provide confidence intervals or error propagation. Our primary aim was to extract a reliable slope from the data for flux calculation, rather than conducting a full statistical analysis of the fit.**

- Figure 2. Instead of plotting wind direction on a y axis that goes from 0 to 359 (this value is cyclical), it should be plotted as an "arrow graph" (see example in Fig. 3 of Puig et al. (2003)). Another alternative would be to choose a cyclic colormap for the wind direction and use it on wind speed data (Fig. 2b).

  In both Fig. 2d and e, there seems to be artifacts in the data (spikes of SST, BW Temp, and BW $O_2$). The O2 concentration obtained from Winkler method in Fig. 2e is not sufficiently visible.

  **We have updated the figure by adding an arrow graph to indicate both wind direction and wind speed. The previously visible spikes were not artifacts but represented the lowering and retrieval of the ROVER. These have now been removed from the figure, making the Winkler-analysis-derived $O_2$ concentration more clearly visible.**

- Finally, to give Fig. 3 a bit more context, consider adding an arrow (or similar marker) above Fig. 2 for each CTD deployment. That way, the reader will not have to find the environmental conditions during each CTD deployment in Fig. 2.

  **We have modified the figure and added an arrow for CTD deployment.**

- Lines 410-416. Add a more detailed description of the differences, or lack of, of POC, PON and CaCO3, as done in lines 417-418 for pyrite contents.

  **We have added the surface values and described the variation between the control (CL) and HI sites. The following sentence has been included in the revised manuscript: 'The surface concentrations of POC, PON, and $CaCO_3$ were consistently higher at the CL site (POC: 3.58 ± 0.74 wt.%; PON: 0.54 ± 0.11 wt.%; $CaCO_3$: 5.39 ± 1.12 wt.%) compared to the HI site (POC: 2.77 ± 0.78 wt.%; PON: 0.41 ± 0.12 wt.%; $CaCO_3$: 3.55 ± 0.48 wt.%).**

- Line 431. Depth-integrated SRR at the control and impact sites should have a measure of uncertainty, no?

  **Yes, the values include uncertainty. We have revised the text to reflect this, now reporting the depth-integrated SRR as 3.9 ± 3.0 and 4.9 ± 2.9 mmol S $m^{-2}$ $d^{-1}$ at the CL and HI sites, respectively.**

- Line 468. Remove "fluxes of" in "showing the fluxes of solutes fluxes across"

  **We have modified the figure caption.**

- Line 490. The authors mention that the two approaches have "slight differences in magnitude for TA and NH4+". Are these differences statistically significant? And the lower H4SiO4 fluxes of BIGO, were they significantly lower?

  **In the case of TA and $NH4^+$, the differences are not statistically significant, whereas for silicate, the difference is statistically significant. Now included in the manuscript.**

- Line 563. Add "it" in "[…] since it will affect the rates of […]"

  **We have corrected the sentence.**

- Line 577. Remove the comma after "as well as, the local"

  **Corrected**

- Line 672. The surface pyrite content in the impact and control site are not statistically significantly different.

  **Correct, but as model results show, only a few percent of ambient pyrite pool needs to be oxidized to account for the TA losses. Now clarified in the text.**

- Line 676. Sentence is missing a verb

  **Corrected.**

---

## Referee Report (RR1)

I thank the authors for taking their time to address all of my comments, which I admit were many. The impressive number of samples and analyses performed for this study is a step forward to understand the biogeochemical effects of mobile demersal fisheries, which is why I have thoroughly revised the response to my first review. I believe that there are still several aspects that are not quite clear and would like the authors to further clarify and address in their manuscript before publication.

I have separated the different comments based on the same numbering I employed in the first review, in order to avoid repetition and to indicate what comment and reply I am referring to. In this review, I only address the aspects that I believe the authors should further clarify. In my opinion, all the other several comments have been addressed.

**1. Type of experimental design**

In my first review, I asked the authors to clarify what type of experimental design they followed, since it was hard to identify based on the methods and results of the manuscript:

- 1. Control-Impact (CI), when an experimentally trawled site is compared with a control site
- 2. Before-After (BA), when the same site is sampled before the disturbance and after the disturbance
- 3. Before-After-Control-Impact (BACI), when (at least) one site is sampled before and after the disturbance (impact site), and (at least) another site is sampled at the same time (before and after) but is not disturbed. This approach also includes collecting additional samples in time after the disturbance to get a better temporal variability.

Clarifying and identifying what type of experimental design this study followed is crucial to then know what type of statistical analysis to employ. The authors clarify that their study follow a CI approach, but then mention that they collected samples in the impact site before the disturbance. Hence, it should be either a BA or a BACI approach, not a CI approach.

In their statistical analysis, the authors aggregate all their "control" (before impact?) and "impact" (after impact?) sites to assess if there are any statistically significant differences in the different parameters they studied. This aggregation is done with samples collected in different periods. The authors argue that they can aggregate this temporal data because "their results [...] show that the difference due to trawling is larger than those caused by natural temporal variability in the CL areas". What statistical analyses did the authors do to arrive to these conclusions? This should be clarified.

I would first perform a linear mixed-effects model that accounts for the disturbance (impact/control) time, and the interaction of both. This would allow the authors to properly assess if natural temporal variability is relevant. If the results of this analysis shows that temporality is not relevant, then it is justifiable to average the data in the figures. If the results of this analysis shows that temporality is actually relevant, then the authors can not average the data in the figures. Moreover, this would imply that temporality should be addressed when assessing the biogeochemical impacts of demersal fisheries, which is seldomly done.

The interpretations and conclusions of this manuscript depend on this, which is why I insist on a proper statistical analysis given the large number of samples collected.

**2. Statistical analyses**

Assuming that the statistical analysis employed is correct, the authors mention that TOU and TA are not significantly different among the impact and control sites. However, in several occasions in both

the manuscript and the reply to my reviews, the authors then argue that there is a decrease in the TA flux in the HI site – this decrease is not statistically significant, right? Please be consistent. This is especially important in this study given the large number of proxies measured.

**8. Alkalinity fluxes and their reasons**

First of all, I want to re-iterate that according to the statistical analysis performed (which should also be revised), there is no statistical difference in the TA fluxes between the HI and CI sites (see my comments above). However, the authors go in great detail to explain that sediment disturbance has an effect on TA fluxes.

Assuming that this is a relevant process, the authors point that this reduction in TA fluxes is due to changes in the sulfate reduction. However, in my previous review I noted that the authors identified that there are no statistically significant changes in sulfate reduction or pyrite content. Isn't this contradictory? Please clarify.

In addition, in my previous review, I asked about the influence of carbonate dissolution. To this, the authors replied: "We do not calculate carbonate dissolution for the HI site since here we are interested in the reduction in TA and DIC following trawling, which we ascribed to a reduction in POC degradation and carbonate dissolution (now mentioned). Denitrification (i.e. NO3 flux) is included in the mass balance (Eq. 8) to calculate carbonate dissolution." I had to re-read these sentences several times since I was confused in the contradiction, which I point below to clarify:

- You don't calculate carbonate dissolution for the HI site because you associate the reduction in TA and DIC due to carbonate dissolution? So carbonate dissolution should be calculated, no?
- You include denitrification in the mass balance to calculate carbonate dissolution. So is carbonate dissolution calculated?

As you can see, it is not clear to me whether carbonate dissolution is calculated. If so, what are the values, and is it relevant? In their reply to my comment 7, the authors mention "Yes, the reduction in TA and DIC may be attributed to a decrease in the rate of POC remineralization and calcite dissolution." As you can see, this is not at all clear to me.

Again, all of this is assuming that the changes in TA is statistically significant, which the authors previously say it is not.

**9. Seafloor-water-air box model**

I still find confusing adding two kind of disturbance (with and without changes in DIC and TA fluxes). A more thorough explanation is needed. This is beyond my area of expertise, and I believe other readers would like to understand this better.

**Lines 325-336. Calculation of diffusion fluxes using FindFit function**

The authors should use a fitting that provides measures of uncertainties within a specific confidence interval to be able to compare the diffusive fluxes across sites and see if they are significant while accounting for their confidence interval.

**Lines 410-416. Description of POC, PON, and CaCO3 variations.**

The authors provided the revised sentence with a description of POC, PON and CaCO3 variations across sites, which is exactly what I meant in my previous comment. However, this revised sentence should also include the results of the statistical analysis which are not statistically significant.

---

## Author Response (AR2)

**Response to reviewer 1**

The authors have addressed most of my concerns, and I am happy to see it published after consideration of a last few comments, listed below.

**Thank you for your positive feedback, Sebastiaan. Please find our detailed response below.**

- 1. The use of Bottom trawling versus mobile-bottom contact fishing. If you acknowledge that it is the more precise term, why would you not just use it? Bottom trawling might be used in other papers, but that is not a reason not to use the correct term.
  - Thank you, we have corrected the terminology throughout.
- 2. The use of 'severe' in the title. Severe is a subjective term, and your paper concludes the impact of the fluxes is not that important. I would urge to remove the term 'severe'. We have modified the title.
- 3. L36: I would add 'potential' to this sentence, as you assumed a rate from a different experiment. So, you have no evidence to say that pyrite oxidation is definitely important at your site.

Added.

- 4. The explanation of the model setup:
  - "In this study, the model was modified in two ways; (i) to include constant values of carbonate dissolution at the derived rates of 9 mmol TA m-2 d-1 and 4.5 mmol DIC m-2 d-1, and (ii) during each trawling event, the benthic DIC and TA fluxes were reduced by 9.2 and 6.7 mol m-2 d-1 in accordance with the data, and for a period of 21 days." This does not make sense to me – the TA flux from the sediment is a consequence of carbonate dissolution. So, do you mean you have a baseline flux of 9 mmol TA m-2 d-1, and reduce this by 6.7 mmol m-2 d-1 (not mol m-2 d-1 I hope?)? If this is the case, please rephrase the description.

If not, then the model setup is probably incorrect and needs to be reconsidered

Thanks for noting the typo in our manuscript (line 695, missing m in front of mol). We corrected this typo and now give the decrease in the TA flux during trawling events in correct units as 6.7 mmol m-2 d-1. We also expanded our model description (line 692 – 695) as follows: ". Benthic DIC fluxes in the original model were exclusively due to POC degradation, whereas TA fluxes were driven by pyrite burial and oxidation. In this study, the model was modified in two ways; (i) to include baseline values of carbonate dissolution at the derived rates of 9.4 mmol TA m-2 d-1 and 4.7 mmol DIC m-2 d-1, and (ii) during each trawling event, the benthic DIC and TA fluxes were reduced from the baseline values by 9.2 and 6.8 mmol m-2 d-1 in accordance with the flux data in Fig. 7, and for a period of 21 days. With this approach, we consider that carbon degradation and carbonate dissolution were

**diminished since most of the degradation and dissolution takes place in surface sediments that were partly removed by trawling."**

**Response to reviewer 2**

I thank the authors for taking their time to address all of my comments, which I admit were many. The impressive number of samples and analyses performed for this study is a step forward to understand the biogeochemical effects of mobile demersal fisheries, which is why I have thoroughly revised the response to my first review. I believe that there are still several aspects that are not quite clear and would like the authors to further clarify and address in their manuscript before publication.

**Thank you, Sarah, for reviewing the manuscript again and providing your comments. Please find our detailed responses to each of them below.**

I have separated the different comments based on the same numbering I employed in the first review, in order to avoid repetition and to indicate what comment and reply I am referring to. In this review, I only address the aspects that I believe the authors should further clarify. In my opinion, all the other several comments have been addressed.

**1. Type of experimental design**

In my first review, I asked the authors to clarify what type of experimental design they followed, since it was hard to identify based on the methods and results of the manuscript:

- 1. Control-Impact (CI), when an experimentally trawled site is compared with a control site
- 2. Before-After (BA), when the same site is sampled before the disturbance and after the disturbance
- 3. Before-After-Control-Impact (BACI), when (at least) one site is sampled before and after the disturbance (impact site), and (at least) another site is sampled at the same time (before and after) but is not disturbed. This approach also includes collecting additional samples in time after the disturbance to get a better temporal variability. Clarifying and identifying what type of experimental design this study followed is crucial to then know what type of statistical analysis to employ.

The authors clarify that their study follow a CI approach, but then mention that they collected samples in the impact site before the disturbance. Hence, it should be either a BA or a BACI approach, not a CI approach.

In their statistical analysis, the authors aggregate all their "control" (before impact?) and "impact" (after impact?) sites to assess if there are any statistically significant differences in the different parameters they studied. This aggregation is done with samples collected in different periods. The authors argue that they can aggregate this temporal data because "their results [...] show that the difference due to trawling is larger than those caused by natural temporal variability in the CL areas". What statistical analyses did the authors do to arrive to these conclusions? This should be clarified.

I would first perform a linear mixed-effects model that accounts for the disturbance (impact/control) time, and the interaction of both. This would allow the authors to properly

assess if natural temporal variability is relevant. If the results of this analysis show that temporality is not relevant, then it is justifiable to average the data in the figures. If the results of this analysis show that temporality is actually relevant, then the authors cannot average the data in the figures. Moreover, this would imply that temporality should be addressed when assessing the biogeochemical impacts of demersal fisheries, which is seldomly done.

The interpretations and conclusions of this manuscript depend on this, which is why I insist on a proper statistical analysis given the large number of samples collected.

We apologize for the confusion. Our study is designed as a control-impact experiment, in which one area was trawled (HI) and the other untrawled (control). Samples were collected from both areas during the study period to assess the impact relative to the control. As mentioned, two MUCs were taken from the HI area before trawling. Two data points does not adequately represent the pre-disturbance condition required for a Before–After (BA) design. For this reason, we stated that the two cores (MUC1 and MUC2) collected from the HI area prior to the trawling experiment were considered control samples.

Regarding the statistical analysis of temporal variability, we had not previously conducted such an analysis. As suggested, we performed a linear mixed-effects model analysis, and the results indicate that TOU, TA, and DIC exhibited temporal variability. In contrast, the remaining parameters (NH4, PO4, SiOH4, NO3, and NO2) did not show significant temporal variability. Please note that the data points shown in the figure (Fig. 8) are not averages but individual measurements.

2. Statistical analyses. Assuming that the statistical analysis employed is correct, the authors mention that TOU and TA are not significantly different among the impact and control sites. However, in several occasions in both the manuscript and the reply to my reviews, the authors then argue that there is a decrease in the TA flux in the HI site – this decrease is not statistically significant, right? Please be consistent. This is especially important in this study given the large number of proxies measured.

We estimated fluxes in two ways: one using landers (total fluxes) and the other based on porewater/diffusive fluxes. The lander fluxes showed significant variability between the control and HI areas, whereas the diffusive flux of TA was not significantly different. For the box model, we only considered the total fluxes measured by the lander, as these results are statistically significant and total fluxes best represent the natural net flux. This was stated in the manuscript: \*'Total fluxes of TA (\*\*p value), DIC (\*p value), NH4+ (\*p value), PO43- (\*p value), and H4SiO4 (\*\*p value) exhibited statistically significant differences between the HI and CL areas.'

8. Alkalinity fluxes and their reasons

First of all, I want to re-iterate that according to the statistical analysis performed (which should also be revised), there is no statistical difference in the TA fluxes between the HI and CI sites (see my comments above). However, the authors go in great detail to explain that sediment disturbance has an effect on TA fluxes. Assuming that this is a relevant process, the authors point that this reduction in TA fluxes is due to changes in the sulfate reduction. However, in my previous review I noted that the authors identified that there are no statistically significant changes in sulfate reduction or pyrite content. Isn't this contradictory? Please clarify. In addition, in my previous review, I asked about the influence of carbonate dissolution. To this, the authors replied: "We do not calculate carbonate dissolution for the HI site since here we are interested in the reduction in TA and DIC following trawling, which we ascribed to a reduction in POC degradation and carbonate dissolution (now mentioned). Denitrification (i.e. NO3 flux) is included in the mass balance (Eq. 8) to calculate carbonate dissolution." I had to re-read these sentences several times since I was confused in the contradiction, which I point below to clarify: - You don't calculate carbonate dissolution for the HI site because you associate the reduction in TA and DIC due to carbonate dissolution? So, carbonate dissolution should be calculated, no? - You include denitrification in the mass balance to calculate carbonate dissolution. So, is carbonate dissolution calculated? As you can see, it is not clear to me whether carbonate dissolution is calculated. If so, what are the values, and is it relevant? In their reply to my comment 7, the authors mention "Yes, the reduction in TA and DIC may be attributed to a decrease in the rate of POC remineralization and calcite dissolution." As you can see, this is not at all clear to me. Again, all of this is assuming that the changes in TA is statistically significant, which the authors previously say it is not.

As noted above, total TA fluxes were significantly different between control and high-impact areas. We did not state that the reduction in TA fluxes is due to changes in sulfate reduction. Regarding carbonate dissolution, we calculated it for the CL area using the TA mass balance in Eq. 8, to provide a baseline estimate of this process for the purpose of including it in the box model. We cannot do this for the HI sites since the sediments were disturbed. Denitrification is included in the mass balance, along with the ammonium flux and pyrite burial. The net result of these fluxes gives carbonate dissolution (RCaDiss, Eq. 8). In the text after Eq. 8, the dissolution rate is provided.

9. Seafloor-water-air box model I still find confusing adding two kind of disturbance (with and without changes in DIC and TA fluxes). A more thorough explanation is needed. This is beyond my area of expertise, and I believe other readers would like to understand this better.

In our experimental study, we observed a reduction in TA and DIC fluxes as a result of trawling (comparing CL and HI areas). To assess whether these reductions influence air—sea CO2 exchange, we applied a previously established model developed by Kalappurakkal et al. (2025) that did not include carbonate dissolution (not clarified). We adapted this model by incorporating our baseline carbonate dissolution rates (i.e., using

data from the control area, Eq. 8). In one simulation run ("standard run"), we reduced the TA and DIC fluxes according to our flux measurements (Fig. 7). In a second run ("No directly imposed TA reduction"), the reduction in TA and DIC fluxes due to trawling was ignored. By comparing these two simulations, the impact of TA and DIC flux reduction on CO2 exchange can be evaluated.

Lines 325-336. Calculation of diffusion fluxes using FindFit function. The authors should use a fitting that provides measures of uncertainties within a specific confidence interval to be able to compare the diffusive fluxes across sites and see if they are significant while accounting for their confidence interval.

We appreciate the suggestion regarding the use of fitting methods that provide uncertainties within a specific confidence interval for diffusive flux estimates. We recognize the importance of illustrating variability across sites, and rather than incorporating formal confidence intervals into the model fitting, we chose to present the variability across replicates for each site by averaging the flux values and visualizing their distribution using box-and-whisker plots. This approach provides a clear, interpretable representation of the spread and central tendency of fluxes across the six datasets, allowing for direct visual comparison of site-to-site variability. We have updated the manuscript to reflect this methodological decision in methods section, and we believe this presentation adequately captures the uncertainty inherent in the measurements for the purpose of inter-site comparison:

"To assess variability in porewater fluxes across datasets and to facilitate comparison with the BIGO fluxes, we computed the average flux per site and visualized the distribution using box-and-whisker plots. This descriptive approach provides a visual representation of the spread in the data, facilitating comparisons between the HI and CL areas."

Lines 410-416. Description of POC, PON, and CaCO3 variations. The authors provided the revised sentence with a description of POC, PON and CaCO3 variations across sites, which is exactly what I meant in my previous comment. However, this revised sentence should also include the results of the statistical analysis which are not statistically significant.

Thank you for pointing this out. We have now included the results for TS and C/N ratios. The revised sentence reads as follows: "In the surface sediment layer, the concentrations of TS and C/N ratios showed no significant differences between the CL (TS:  $0.62 \pm 0.13$  wt.%; C/N:  $7.75 \pm 0.26$ ) and HI (TS:  $0.49 \pm 0.17$  wt.%; C/N:  $7.93 \pm 0.38$ ) sites."